# *Streptococcus pyogenes* Forms Serotype- and Local Environment-Dependent Interspecies Protein Complexes

Sounak Chowdhury,[a] Hamed Khakzad,[b,c] Gizem Ertürk Bergdahl,[a] Rolf Lood,[a] Simon Ekstrom,[d] Dirk Linke,[e] Lars Malmström,[a] Lotta Happonen,[a] Johan Malmström[a,d]

[a]Division of Infection Medicine, Department of Clinical Sciences, Faculty of Medicine, Lund University, Lund, Sweden
[b]Equipe Signalisation Calcique et Infections Microbiennes, Ecole Normale Supérieure Paris-Saclay, Gif-sur-Yvette, France
[c]Institut National de la Santé et de la Recherche Médicale U1282, Gif-sur-Yvette, France
[d]BioMS, Lund, Sweden
[e]Department of Biosciences, Section for Genetics and Evolutionary Biology, University of Oslo, Oslo, Norway

**ABSTRACT** *Streptococcus pyogenes* is known to cause both mucosal and systemic infections in humans. In this study, we used a combination of quantitative and structural mass spectrometry techniques to determine the composition and structure of the interaction network formed between human plasma proteins and the surfaces of different *S. pyogenes* serotypes. Quantitative network analysis revealed that *S. pyogenes* forms serotype-specific interaction networks that are highly dependent on the domain arrangement of the surface-attached M protein. Subsequent structural mass spectrometry analysis and computational modeling of one of the M proteins, M28, revealed that the network structure changes across different host microenvironments. We report that M28 binds secretory IgA via two separate binding sites with high affinity in saliva. During vascular leakage mimicked by increasing plasma concentrations in saliva, the binding of secretory IgA was replaced by the binding of monomeric IgA and C4b-binding protein (C4BP). This indicates that an upsurge of C4BP in the local microenvironment due to damage to the mucosal membrane drives the binding of C4BP and monomeric IgA to M28. These results suggest that *S. pyogenes* has evolved to form microenvironment-dependent host-pathogen protein complexes to combat human immune surveillance during both mucosal and systemic infections.

**IMPORTANCE** *Streptococcus pyogenes* (group A *Streptococcus* [GAS]), is a human-specific Gram-positive bacterium. Each year, the bacterium affects 700 million people globally, leading to 160,000 deaths. The clinical manifestations of *S. pyogenes* are diverse, ranging from mild and common infections like tonsillitis and impetigo to life-threatening systemic conditions such as sepsis and necrotizing fasciitis. *S. pyogenes* expresses multiple virulence factors on its surface to localize and initiate infections in humans. Among all these expressed virulence factors, the M protein is the most important antigen. In this study, we perform an in-depth characterization of the human protein interactions formed around one of the foremost human pathogens. This strategy allowed us to decipher the protein interaction networks around different *S. pyogenes* strains on a global scale and to compare and visualize how such interactions are mediated by M proteins.

**KEYWORDS** DIA-MS, host-pathogen interactions, M proteins, protein-protein interactions, *Streptococcus pyogenes*, XL-MS

Address correspondence to Johan Malmström, Johan.Malmstrom@med.lu.se.

Bacterial pathogens have evolved to express a multitude of virulence factors on their surface to establish versatile host-pathogen protein-protein interactions (HP-PPIs) (1). These interactions range from binary interactions between two proteins to the formation of multimeric interspecies protein complexes that enable bacterial

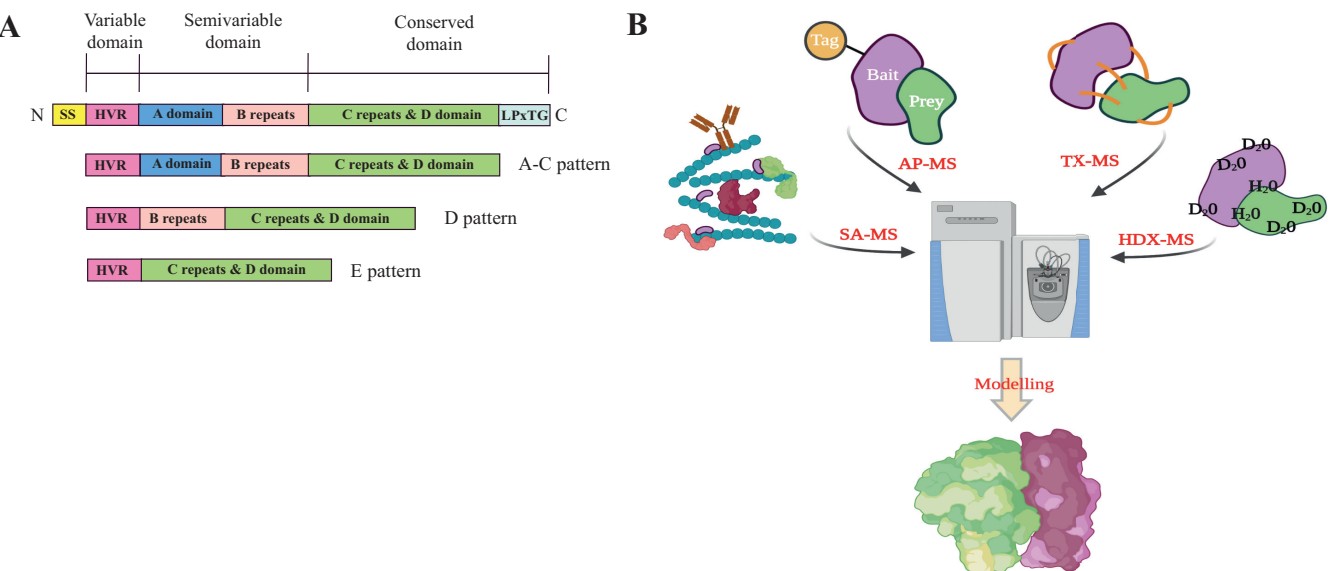

**FIG 1** M protein (naive and mature) structure and experimental overview to identify *S. pyogenes*-human protein interactions. (A) Arrangement of the different domains in M proteins. SS, signal sequence; HVR, hypervariable region domain, which is unique among different M proteins, thereby giving rise to the numerous *S. pyogenes* serotypes. A domain and B repeats form the semivariable domain; C repeats and D domains, including the LPXTG anchor sequence, form the conserved domain. Cleavage of the SS leads to a mature M protein, and LPXTG helps the M protein anchor the bacterial surface. The S regions of certain M proteins are not represented. M proteins are classified into the A-C pattern harboring the A domain, B and C repeats, and the D domains; the D pattern comprising the B-C repeats and the D domains; and the E type harboring only C repeats and the D domain. The M proteins are not drawn to scale. (B) Schematic overview of the integrative approach used to characterize the protein network and complex around *S. pyogenes*. In SA-MS, pathogens are incubated with complex biological mixtures to capture proteins interacting with the bacterial surface, which are then identified and quantified by MS. In AP-MS, recombinant bait proteins are expressed, which are then made to capture prey proteins from complex biological mixtures, followed by identification and quantification by MS. TX-MS is used to cross-link the protein partners to map the binding site, while HDX-MS identifies the binding site based on the exchange of $H_2O$ and $D_2O$. Computational modeling is then used to generate protein interaction models based on the identified protein interaction sites. (The illustration in panel B was created using BioRender.)

pathogens to hijack and rewire molecular host systems to circumvent immune defenses. One prominent example is *Streptococcus pyogenes* (group A *Streptococcus* [GAS]), a Gram-positive and beta-hemolytic bacterium. This bacterium causes diverse clinical manifestations such as mild and local infections like tonsillitis, impetigo, and erysipelas as well as life-threatening systemic diseases like sepsis, meningitis, and necrotizing fasciitis (2). Globally, 700 million people suffer from *S. pyogenes* infections every year, leading to an estimated 160,000 deaths (3), thus making *S. pyogenes* a widespread bacterial pathogen in the human population. *S. pyogenes* abundantly produces a prominent surface antigen, the M protein, known to enable bacterial invasion into human cells, prevent phagocytosis (4, 5), and promote survival in infected tissues (6, 7). These M proteins are dimeric α-helically coiled-coil proteins covalently attached to the *S. pyogenes* cell wall and extending approximately 500 Å into the extrabacterial space to form a dense fibrillary coat on the bacterial surface (8). The M proteins consist of several protein domains, some of which are repeat regions (Fig. 1A). The N-terminal 50 amino acid (aa) residues constitute the hypervariable region (HVR) (9, 10). Sequence variation within the HVR is used to classify the M protein, and to date, >220 distinct *S. pyogenes* serotypes have been reported (8). The HVR is followed by a stretch of 100 to 150 amino acids that forms the semivariable domain of the M proteins and encompasses the A domain and the B repeats. The subsequent C repeats and the D domain form the conserved C-terminal part of the M proteins. Based on the domain arrangement of the M proteins and the presence of *emm* and *emm*-like genes in the GAS genome, M proteins are classified into different *emm* patterns, e.g., A-C, D, and E (11, 12). *emm* pattern A-C represents long M proteins with A, B, C, and D domains, and *emm* pattern D includes M proteins with B, C, and D domains, while *emm* pattern E includes only the C and D domains (12, 13) (Fig. 1A). It has been reported that *emm* pattern A-C mainly includes *S. pyogenes*

strains associated with throat infections and that *emm* pattern D includes *S. pyogenes* strains responsible for skin infections, while the E pattern includes generalist *S. pyogenes* strains typically infecting both sites (13), indicating that the M protein domain composition correlates with host tissue tropisms. Furthermore, comparative sequence analysis of the M proteins enables the classification of the M proteins into clades. Clade X includes the E pattern, and clade Y includes the A-C pattern, while pattern D seems to fall into both clades X and Y (14).

The diverse domain arrangement and partially high sequence variability of the M proteins enable *S. pyogenes* to form protein interactions with various human proteins (15–17). A recent chemical cross-linking mass spectrometry (MS) and structural modeling study showed that the M1 protein of *emm* pattern A-C is capable of forming a large 1.8-MDa interspecies protein complex with up to 10 different human proteins (18). The model by Hauri et al. shows that the interacting human proteins are precisely placed along the $\alpha$-helically coiled-coil structure of the M1 protein. In this way, the M protein can form a highly organized human plasma protein interaction network on the bacterial surface consisting of both human-human and *S. pyogenes*-human protein interactions (19). One example is the binding of fibrinogen to the B repeats of the M protein (10, 15, 20), where fibrinogen in turn mediates binding to factor 13 (F13). Fibrinogen binding to the M protein prevents the deposition of opsonizing antibodies to inhibit phagocytosis (21–23). Several copies of human serum albumin (HSA) have been proposed to bind the C repeats of the M proteins to facilitate the uptake of fatty acids and promote growth during stationary phase (15, 21, 24, 25). Additionally, certain *S. pyogenes* serotypes can bind immunoglobulin G (IgG). The orientation of IgG binding, i.e., whether it is Fab or Fc mediated, is governed by the concentration of IgGs in the host niche (16, 26). Binding to IgG Fc is mediated by the S region found in some M proteins and located between the B and C repeats and the HVR (15, 27). Other M proteins such as M4 and M22 of *emm* pattern E have been shown to bind immunoglobulin A (IgA) (16, 28). This binding occurs between the N terminus of M proteins (29, 30) and the interdomain region of IgA Fc, which is also the known binding site of the human IgA receptor CD89 (31). The binding of M proteins to IgA Fc blocks the binding of IgA to CD89, thus preventing IgA effector functions, inhibiting phagocytosis, and promoting bacterial virulence (31, 32). In addition, many different *emm* pattern M proteins bind complement system C4b-binding protein (C4BP) to the N-terminal HVR domain (33–37). C4BP bound to the M protein sequesters C4b from plasma and acts as a cofactor for the degradation of C4b by complement factor I (34, 38, 39), thereby inhibiting the complement pathway and phagocytosis of the bacterium (32).

Collectively, these previous studies indicate that the domain arrangement of different M proteins impacts the HP-PPI networks that are formed around the streptococcal surface. However, the large variability between different M proteins, the difficulty in pinpointing exact binding interfaces, and the formation of human-human protein interactions at the streptococcal surface make it challenging to determine the structure and composition of such interspecies protein networks. A more detailed understanding of how the domain arrangement determines the composition of M protein-centered interspecies proteins complexes could help explain differences in tissue tropism observed between different *emm* types. Here, we applied quantitative and structural mass spectrometry techniques in an unbiased fashion to show that different serotypes form highly distinct *emm* pattern-specific HP-PPI networks. These interaction networks depend to a large extent on the type of M proteins expressed by a given strain. Furthermore, by in-depth structural mass spectrometry and structural modeling analyses, we demonstrate that the M proteins form different protein complexes depending on the local microenvironments to facilitate immune evasion strategies in different ecological niches.

## RESULTS

**Human plasma protein interaction networks with *S. pyogenes* surface proteins.**
M proteins are long extended surface-attached proteins with various combinations of

A, B, C, and D domains (Fig. 1A) that allow the M proteins to engage in numerous protein interactions simultaneously. While the protein interaction network formed around A-C patterns is relatively well described (17), less is known about the protein interaction network organized around E pattern strains. Here, we combined quantitative and structural mass spectrometry techniques to determine how the different M protein domains within the E and A-C patterns influence the composition and structure of the human plasma-*S. pyogenes* interaction network (Fig. 1B). First, we selected three representative clinical isolates from *emm* pattern type A-C (M1, M3, and M5) and three from type E (M28, M49, and M89) and performed bacterial surface adsorption mass spectrometry (SA-MS) analysis (17), as schematically shown in Fig. 1B. In this analysis, the clinical isolates were incubated with pooled normal human plasma. Surface-adhered proteins were enriched via centrifugation and quantified by data-independent mass spectrometry analysis (DIA-MS). The data were stringently filtered, resulting in the identification of 92 surface-bound plasma proteins in total, which were further grouped into six major protein families according to their functional roles, i.e., apolipoproteins, cell adhesion and cytoskeleton proteins, coagulation, complement, immunoglobulins, and other plasma proteins (Fig. 2A; see also Table S3 in the supplemental material). The clustering of the strains in the heat map is based on the Z-score, where the Z-score measures the standard deviation of a protein intensity from the mean intensity of that protein across all strains. The quantitative data matrix across each strain shows that there are marked differences in the HP-PPI networks formed on the streptococcal surface between the serotypes producing A-C- and those producing E-type M proteins (Fig. 2A). The A-C pattern strains typically bind fibrinogen and components of the complement system, whereas the E pattern strains bind with various apolipoproteins, immunoglobulins, and components from the complement and coagulation system, such as C4BP and vitamin K-dependent protein S (PROS) (Fig. 2A). The analyzed strains were furthermore capable of forming distinct serotype-specific interaction networks, also within their respective A-C or E patterns. To objectively determine the major components of these networks, we used coexpression network analysis (Fig. 2B). This analysis revealed four highly connected protein clusters (highlighted using semitransparent ellipses) of strongly correlating human proteins (blue lines) ($r^2 > 0.9$) that bind to one or two of the strains. The highest-correlating proteins associated with M49 and M89 networks, indicated by the thickness of the lines, were several apolipoproteins such as APOH and APOC4 (Fig. 2B). In contrast, M3, M5, and, to some degree, M1 bound fibrinogen, whereas M28 predominately associated with several proteins such as C4BP, PROS, APOB, and IgA (Fig. 2B). Interestingly, the network view also shows that there are several proteins that are negatively correlated ($r^2 < -0.6$), as indicated by the red lines in Fig. 2B. To further visualize these binding patterns, correlation plots were plotted for selected protein pairs from each protein cluster (Fig. 2C). As expected, strong correlations were observed between proteins belonging to the same protein cluster in Fig. 2B, such as APOH-APOC4, FIBB-FIBA, C4BPA-C4BPB, APOB-PROS, IGHA1-IGHA2, CO3-CO4A, C4BPA-PROS, and IGHA1-C4BPA. In contrast, other proteins appear to bind significantly more to some strains, such as fibrinogen and C4BP, where M1, M3, and M5 bind fibrinogen but not C4BP, while M28, M49, and M89 bind C4BP but not fibrinogen (Fig. 2C). Moreover, high levels of fibrinogen were related to low levels of several other proteins of the M28, M48, and M89 networks, such as IgA, PROS, and components of apolipoproteins such as APOH (Fig. 2C). These results demonstrate that each strain can assemble strain-specific HP-PPI networks and that there are substantial differences in the interaction networks between *emm* types.

**Human plasma protein interaction networks with *S. pyogenes*-M proteins.** To understand to what degree differences in the domain arrangements of the M proteins (Fig. 1A) mediate the differential patterns of binding of human proteins to the *S. pyogenes* strains, we applied protein affinity purification mass spectrometry (AP-MS) (17), as schematically shown in Fig. 1B. Based on the different *S. pyogenes* strains screened as described above, six M proteins (M1, M3, M5, M28, M49, and M89) were recombinantly expressed

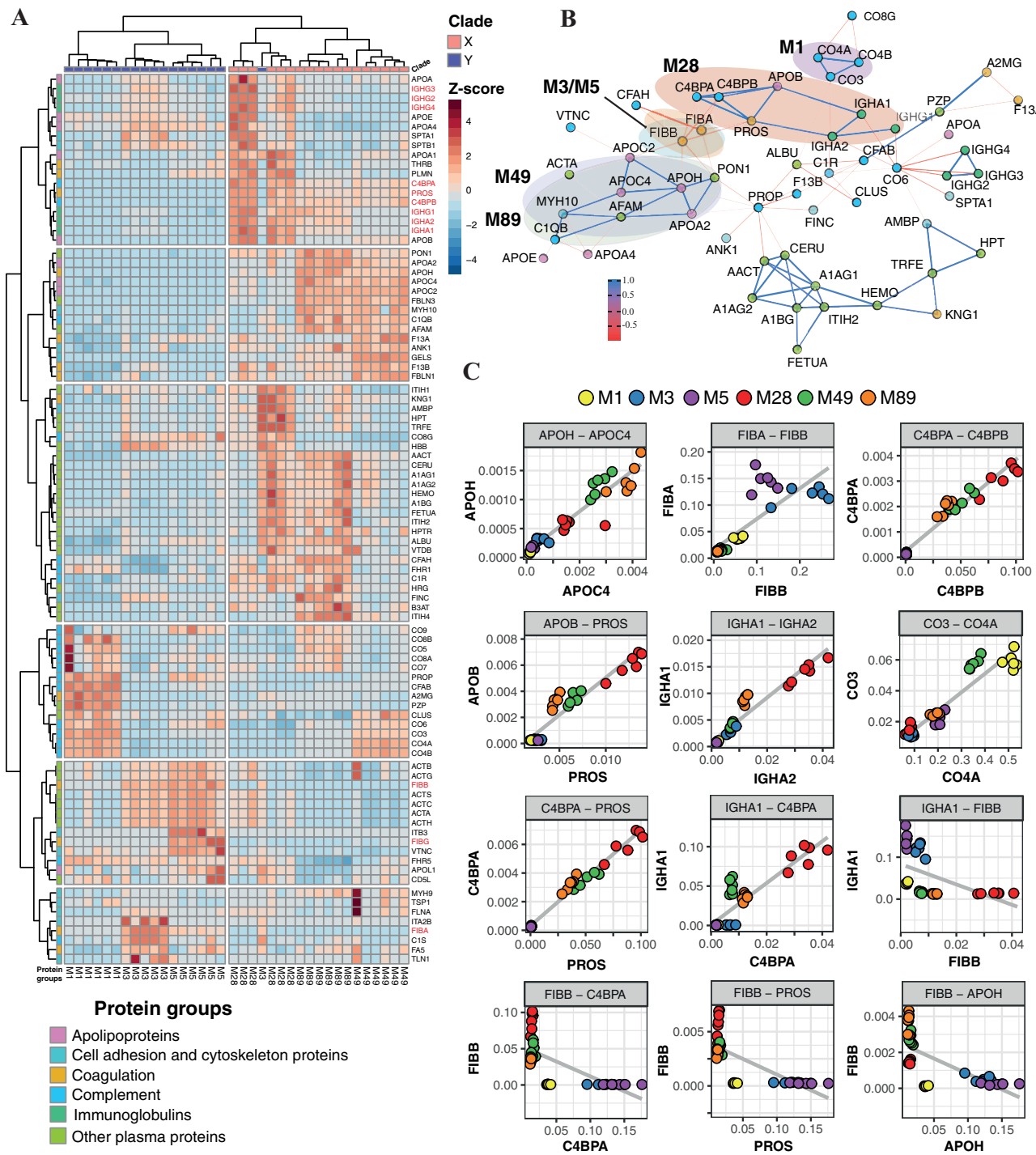

**FIG 2** Human plasma proteins interacting with different *S. pyogenes* serotypes identified by SA-MS. (A) Cluster analysis of 92 human plasma proteins interacting with six different *S. pyogenes* serotypes, M1, M3, and M5 (clade Y) and M28, M49, and M89 (clade X). SA-MS data are represented for 6 independent replicates for each strain. Human plasma proteins were categorized into six protein families, i.e., apolipoproteins, cell adhesion and cytoskeleton proteins, coagulation, complement, immunoglobulins, and other plasma proteins. The proteins in red are discussed in the text. One of the M3 replicates is an outlier, thereby falling in the X clade. A Z-score value of $-4$ to $+4$ represents the standard deviation from the mean protein intensity (zero). Red represents highly enriched proteins, while blue represents less abundant proteins. (B) Pearson correlation network analysis of human plasma proteins across six different serotypes. Colored ellipses are drawn around each M serotype to represent protein groups enriched in the individual serotypes. Each protein is depicted by a circle and colored according to the protein family to which it belongs. Proteins present in the ellipse bind strongly to particular serotypes. The ellipses of M3-M5 and M49-M89 are overlapping as they bind to common proteins, while M1 and M28 have nonoverlapping ellipses. Proteins outside the ellipse could not be confidently clustered in a particular serotype as they bound to most of the strains. Blue lines represent strongly correlating proteins ($r^2 > 0.9$), while red lines represent mutually exclusive proteins or negatively correlating proteins ($r^2 < -0.6$). The thickness of the line represents how strongly the proteins are correlated. (C) Correlation plots for some representative proteins from each protein cluster across different strains. Strains are represented by dots with different colors. The x and y axes represent the protein intensities.

with an affinity tag. The tagged M proteins were used to affinity purify interacting plasma proteins, followed by MS analysis and filtering. Only proteins enriched $>1$ $\log_2$ times (2-fold) and having an adjusted statistical $P$ value of 0.05 compared to green fluorescent protein (GFP)-enriched proteins were considered interactors (see Fig. 3A for an example; see also Fig. S2 in the supplemental material). This filtering strategy generated a final list of 32 high-confidence nonredundant interactions with M1, M3, M28, M49, and M89, categorized into the same functional categories as the ones described above. As M5 had poor protein stability and yield, it could not be used for AP-MS and was excluded from the study. The heat map of the significant interactions in Fig. 3B reveals five predominant column clusters and again demonstrates that the M proteins are involved in distinct protein interactions with human plasma proteins. Fibrinogen binding was prominent with M proteins of *emm* type A-C (M1 and M3), and C4BP binding to M proteins was prominent with *emm* type E (M28, M49, and M89) (Fig. 3B), in a similar fashion as observed in the SA-MS results described above and as previously suggested by Sanderson et al. (14). High levels of fibrinogen were detected to bind M1 by AP-MS than by SA-MS due to the low copy number of the M1 protein in SF370 (data not shown). To detail the properties of the differential binding patterns, we constructed another correlation network plot for the 32 proteins across the five different M proteins (Fig. 3C). Similar to the SA-MS results, the network view shows several correlating protein clusters ($r^2 \geq 0.5$) typically associated with one or two of the analyzed M proteins. Several of the serotype-specific proteins described above, such as fibrinogen, apolipoproteins, C4BP, and IgA, are strongly associated with particular M proteins. In addition, we can confirm that the binding of some proteins seems to result in lower binding of other proteins ($r^2 < -0.5$) such as fibrinogen-PROS and fibrinogen-C4BP$\alpha$, demonstrating that the interactions captured as described above using SA-MS are to a large degree mediated by the M proteins. To visualize the core interaction network between the analyzed M proteins, we selected the highly enriched protein interactions ($\log_2$ enrichment of $>3$ compared to GFP) to plot a schematic interaction network graph (Fig. 3D). The network graph reveals that albumin, IgG1, and IgG4 are equally associated with all analyzed M proteins. Albumin is known to bind the conserved C repeats (15, 21, 24, 25) of the M protein, thus making the association of albumin with all M proteins logical. In addition, IgA2 is enriched in all M proteins but significantly more enriched in M28, which is also coupled to C4BP$\alpha$, IgA1, alpha-1-antitrypsin (A1AT), and, to a lesser degree, PROS. The cysteine residue on the C terminus of the $\alpha$ chain of monomeric IgA has been shown to form disulfide bonds with A1AT (40), and C4BP is known to form a complex with PROS (41). We speculate that these proteins form a larger complex mediated by human-human protein interactions on M28. In contrast, M1 and M3 typically bind fibrinogen and fibronectin, whereas M49 binds several components of the complement system, and both M49 and M89 bind PROS. In conclusion, the results from the AP-MS analysis demonstrate that M proteins play a major role in shaping the serotype-specific HP-PPI networks observed by SA-MS. Although the E-type M proteins are substantially smaller than the A-C-type proteins, their interaction networks with human plasma proteins are still surprisingly complex. To visualize how these short E-type M proteins form interspecies protein complexes, we selected M28 for further structural characterization, with a particular focus on the binding with IgA and C4BP, as outlined in Fig. 1B.

**Characterization of the M28 IgA-C4BP interaction in different local microenvironments.** As we observed that IgA was significantly enriched on M28, we measured the affinity between M28 and IgA by using surface plasmon resonance (SPR). The binding of M28-IgA was compared to M1-IgA binding, which, according to our observations, showed very low or no IgA binding. We immobilized the M proteins (ligands) on the sensor chip and injected IgA (analyte) over them. The kinetic analysis showed the best fit to a heterogeneous ligand model compared to a 1-to-1 model (Fig. S3A, B, and D). Surface heterogeneity (heterogeneous ligand) models are observed if the ligand has multiple binding sites for an analyte. Thus, an explanation for the deviations from a 1-to-1 fitting model could be that IgA has multiple binding sites on M28. The calculated affinity constants showed that IgA had a 3-log-higher affinity for M28 than for M1 (equilibrium

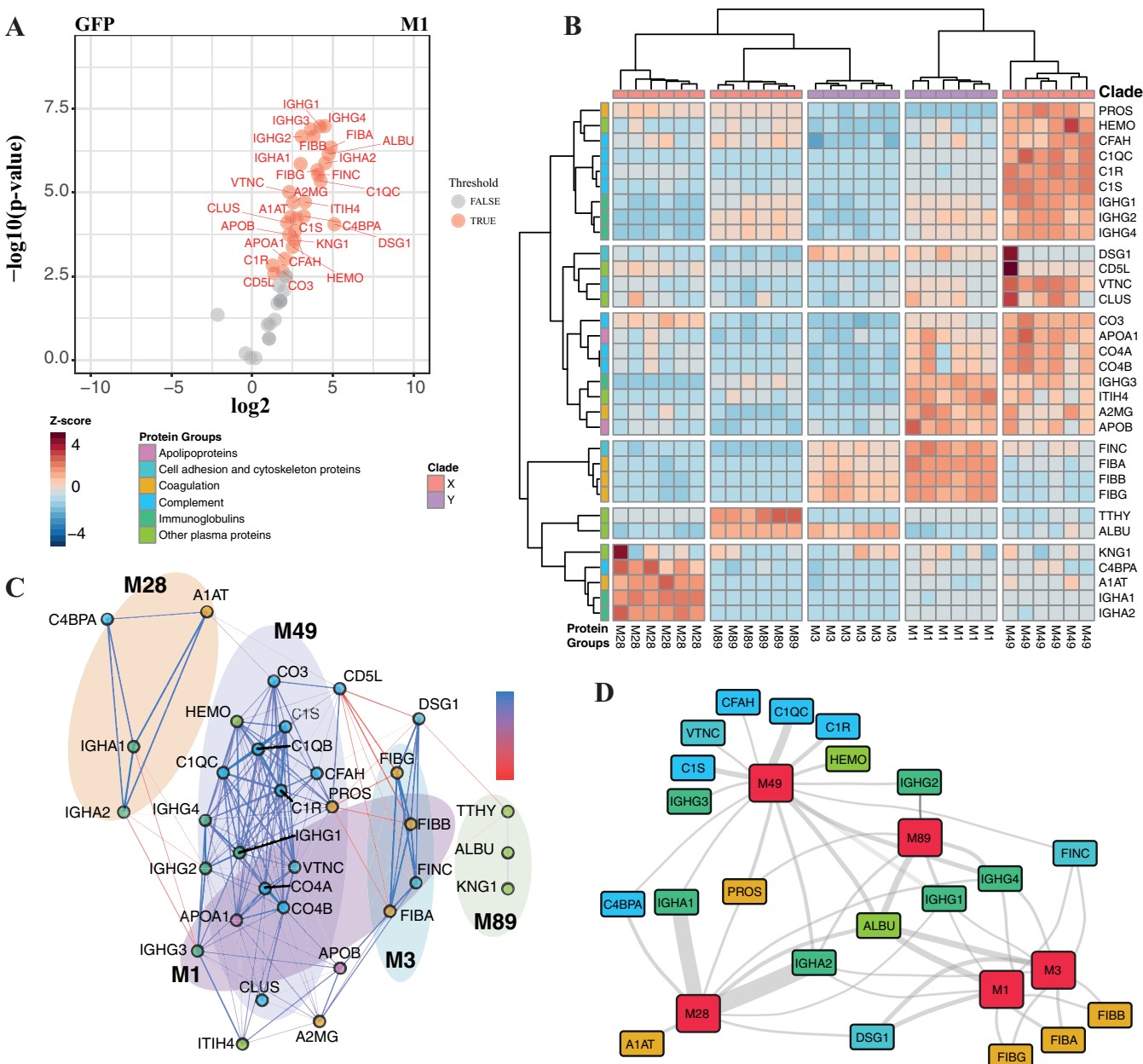

**FIG 3** Human plasma proteins interacting with different M proteins identified by AP-MS. (A) Volcano plot analysis of M1 with GFP. The DIA data were filtered against sfGFP using a $\log_2$ fold enrichment of >1 with an adjusted *P* value of 0.05 using the Student *t* test. The red dots represent high-confidence interactors, while the gray dots represent interactors that did not pass the above-mentioned filtering criteria. Volcano plots for other M proteins and GFP are provided in Fig. S2 in the supplemental material. For clarity, GFP and M proteins are removed from the volcano plot. (B) Cluster analysis of 32 human plasma proteins across five different M proteins for 6 replicates. A Z-score value of −4 to +4 represents the standard deviation from the mean protein intensity (zero). Red represents highly enriched proteins, while blue represents less abundant proteins. (C) Pearson network analysis for $r^2$ values of ≥0.5 for 32 proteins across five M proteins. Each sphere represents a protein cluster. Blue lines represent positively correlated proteins, while red lines represent mutually exclusive ones. (D) Network analysis of 21 highly significant human plasma proteins across five different M proteins. These 21 proteins were >3 $\log_2$ fold enriched in M proteins compared to GFP. The thickness of the line represents the fold change compared to GFP.

dissociation constant 1 [$K_{D1}$] of ~$10^{-10}$ M and $K_{D2}$ of ~$10^{-8}$ M for M28 and $K_{D1}$ of ~$10^{-7}$ M and $K_{D2}$ of ~$10^{-7}$ M for M1) (Fig. 4AI and II). The differences in the $K_{D1}$ and $K_{D2}$ values of IgA toward M28 support two different binding sites on M28 for IgA, one with high affinity and the other with lower affinity. We also performed SPR analysis of the interaction with C4BP and M28 since C4BP was significantly enriched on M28 in our SA-MS and AP-MS experiments described above. As C4BP has 7 $\alpha$ chains and M28 is a dimer, it is likely that we characterize the interaction in terms of avidity. The kinetic analysis for this interaction showed better fitting to a 1-to-1 model (Fig. S3C and D).

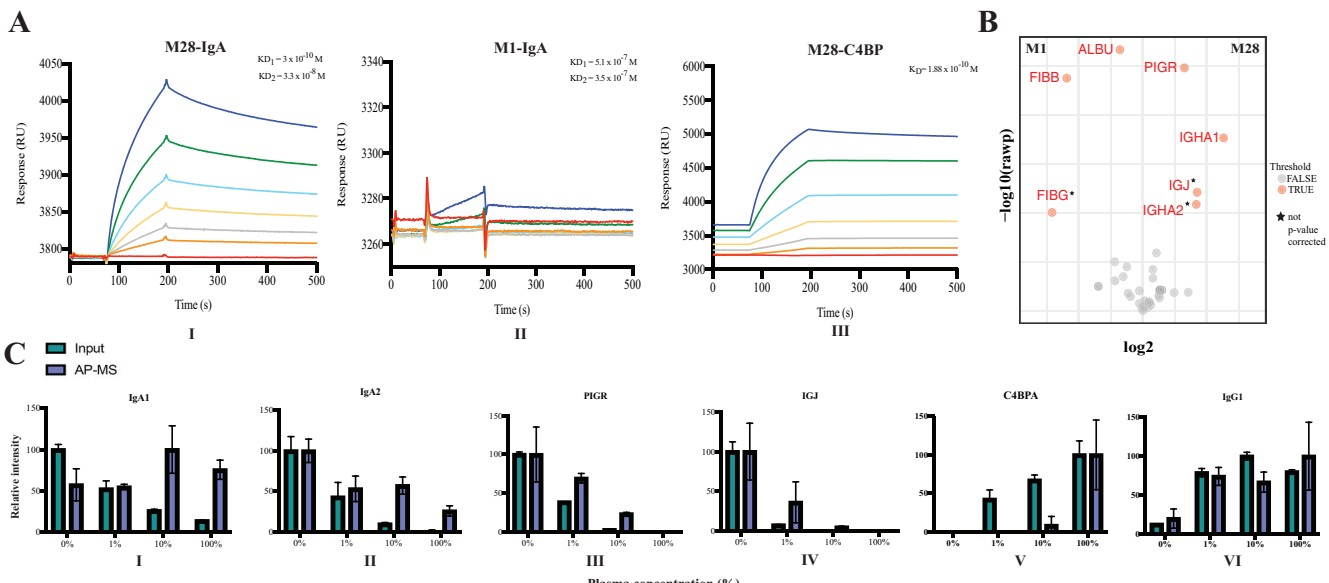

**FIG 4** M protein saliva-plasma interaction. (A) Sensorgrams showing response units (RU) ($y$ axis) plotted as a function of time (in seconds) ($x$ axis) for the interactions of M28 with IgA ($K_{D1} = 3 \times 10^{-10}$ and $K_{D2} = 3.38 \times 10^{-8}$) (I), M1 with IgA ($K_{D1} = 5.13 \times 10^{-7}$ and $K_{D2} = 3.57 \times 10^{-7}$) (II), and M28 with C4BP ($K_D$ of $1.88 \times 10^{-10}$) (III). The different colors of the lines in the sensorgrams represent different concentrations of IgA (red, 0 $\mu$M; orange, 0.009375 $\mu$M; gray, 0.01875 $\mu$M; yellow, 0.0375 $\mu$M; light blue, 0.075 $\mu$M; green, 0.15 $\mu$M; dark blue, 0.3 $\mu$M) and C4BP (red, 0 nM; orange, 3 nM; gray, 6 nM; yellow, 12 nM; light blue, 24 nM; green, 48 nM; dark blue, 96 nM). (B) Volcano plot for AP-MS of saliva with M1 and M28 to identify true human saliva proteins interacting with M28. The data were filtered using a log$_2$ fold enrichment of >1 with an adjusted $P$ value of 0.05 using the Student $t$ test. Proteins marked with stars have nonadjusted $P$ values. Red dots represent high-confidence interactors, while gray dots represent interactors that did not pass the filtering criterion. (C) To understand the levels of proteins enriched on M28 in different dilutions, the data are represented as a comparison between the input and AP-MS. The relative intensity of peptides ($y$ axis) was plotted against 0, 1, 10, and 100% plasma concentrations ($x$ axis), mimicking vascular leakage for IgA1 (I), IgA2 (II), PIGR (III), IGJ (IV), C4BPA (V), and IgG1 (VI). The relative protein intensity (input or pulldown) reflects the ratio of the individual intensity to the mean of the highest protein intensity across all the dilutions expressed as a percentage. Green represents input samples, and purple represents AP-MS with M28.

Affinity constants calculated from the sensorgrams resulted in a $K_D$ of $1.88 \times 10^{-10}$, suggesting a single binding site with a high affinity between M28 and C4BP (Fig. 4AIII).

Most IgA produced in the human body is secreted into the mucous membrane, thereby acting as a first line of defense against infections (28). To understand if an IgA-rich microenvironment changes the *S. pyogenes* M28 protein network, we quantified the protein interaction networks of M1 and M28 in pooled normal human saliva by AP-MS. These experiments showed that IgA binding from saliva occurs only on M28 but not on M1 (Fig. 4B). Additionally, we observed coenrichment between IgA and polymeric immunoglobulin receptor (PIGR) and immunoglobulin J chain (IGJ) (Fig. 4B). PIGR is known to bind polymeric IgA and IgM at the basolateral surface of epithelial cells. PIGR-bound polymeric IgA undergoes transcytosis to the luminal surface, where cleavage by one or more proteinases results in secretory IgA (sIgA) (40). The J chain forms a disulfide bridge between the cysteine residues of the IgA heavy chain, giving rise to multimeric IgA (40).

Polymeric IgA is known to be prevalent in saliva, while monomeric IgA and C4BP are predominantly present in plasma. *S. pyogenes* typically induces vascular leakage when localized in the upper respiratory tract (42, 43). To understand how *S. pyogenes* M28 has adapted to changing microenvironments, we quantified the protein interactions of M28 in a mixture of saliva and plasma. These AP-MS experiments were performed using 100% saliva, 1% plasma in saliva, 10% plasma in saliva, and 100% plasma to mimic conditions during a local infection followed by a systemic infection. The results show that IgA1 binds at similar levels to M28 across all saliva or plasma mixtures, although the concentration of IgA1 is lower in plasma (Fig. 4CI). In contrast, IgA2 binding to M28 predominantly occurs in saliva and decreases with decreasing IgA2 concentrations in plasma (Fig. 4CII). The levels of PIGR and IGJ binding to M28 (Fig. 4CIII and IV) follows a similar trend, although we note proportionally higher levels of these two proteins enriched

on M28 in 10% plasma than the input concentration (Fig. 4CIII and IV). These results imply that M28 can bind IgA in both the sIgA and monomeric forms, where the former is pronounced in saliva. The high levels of affinity-purified IGJ and PIGR compared to the input pool in the 10% plasma environment, with a nearly 18-times-higher plasma protein concentration than in saliva, suggest that the sIgA binds with high affinity, which is in contrast to previously published results (44). The mixed saliva-plasma enrichment comparison of M28 additionally revealed elevated levels of C4BP on M28 only at high plasma concentrations (Fig. 4CV). Interestingly, although there were detectable levels of C4BPA in 1% plasma, there was no strong enrichment of C4BPA with M28 at this low plasma concentration. These results are surprising as the SPR analysis showed that the affinity between M28 and C4BPA was in the subnanomolar range. Possibly, this could be accounted for by the fact that the levels of secretory IgA were still high with 1% plasma. The levels of IgG1 enriched on M28 seemed to increase with increasing plasma concentrations (Fig. 4CVI). Collectively, these results show that M28 binds secretory IgA in saliva and in plasma M28 binds IgA and C4BP.

**Structural determination of M28 with IgA and C4BP.** To understand how the shorter E-type M28 binds secretory IgA in saliva and monomeric IgA and C4BP in plasma, we used two orthogonal structural mass spectrometry techniques, targeted cross-linking mass spectrometry (TX-MS) (18) and hydrogen-deuterium mass spectrometry (HDX-MS). TX-MS involves the use of cross-linking mass spectrometry and computational models to determine protein-binding interfaces. In HDX-MS, the protein complex is subjected to deuterated water, and protected regions, such as protein-binding interfaces, are tracked using MS. The site engaged in binding will block the deuterium exchange, which is then used to map the binding interface. As the input, we first generated a computational model of full-length M28, which was determined using the Rosetta comparative modeling (RosettaCM) protocol (45), based on the previously reported model of the M1 protein (27). This model was further used to provide protein-protein docking decoys using structures deposited in the Protein Data Bank for IgA (PDB accession number 6LXW) and C4BP (PDB accession number 5HYP). For M28, the affinity tag was removed, and the untagged protein was mixed in solution with either C4BP or the Fc domain of IgA, followed by TX-MS and HDX-MS analyses.

For C4BP, we found one interaction site supported by both TX-MS and HDX-MS. The cross-linked peptides observed between M28 and C4BP overlapped the interaction interface resolved using X-ray crystallography (46) (Fig. 5A and BI, Fig. S4, and Table S2). The cross-links were observed between two C4BP residues (K28 and K67 under PDB accession number 5HYP; the corresponding residues are K72 and K111 in the full-length C4BPα chain) and K50 on our M28 construct (Fig. 5A and BI and Table S1). No cross-links from C4BP to the crystallized M28 segment were observed (Fig. 5A and BI), most likely due to the lack of stereochemically favorable lysine residues at the interaction interface. Using HDX-MS, we identified a 14-amino-acid stretch (aa 23 to 36) in the HVR domain of M28 interacting with C4BP (Fig. 5A and BII), enclosed between the C4BP-binding site in the crystallized complex (PDB accession number 5HYP) and the cross-linked site (K50) identified by TX-MS (Fig. 5A).

For IgA, we identified two different cross-linked sites by TX-MS, and similarly, HDX-MS analysis of the M28-IgA Fc domain interaction showed strong protection against deuterium uptake at the same two distinct sites. The first site was supported by four interprotein cross-links and overlaps the previously identified M22-based streptococcal IgA-binding peptide (SAP) (28). (Fig. 5A and CI, Fig. S4, and Table S2). At a 1:1 ratio of M28 to IgA, the reduction in deuterium uptake was observed at the SAP and the overlapping region, as identified by TX-MS (Fig. 5A and CII). In addition, TX-MS also identified a novel IgA Fc interface in the middle of M28 supported by eight high-confidence interprotein cross-links (Fig. 5DI, Fig. S4, and Table S2). A reduction in deuterium uptake was furthermore also observed for residues 112 to 128 at an M28-to-IgA ratio of 2:1, especially at short labeling times (Fig. 5A and DII). Additional AP-MS experiments confirmed that this site is sufficient for enriching IgA but to a lesser extent than the SAP sequence

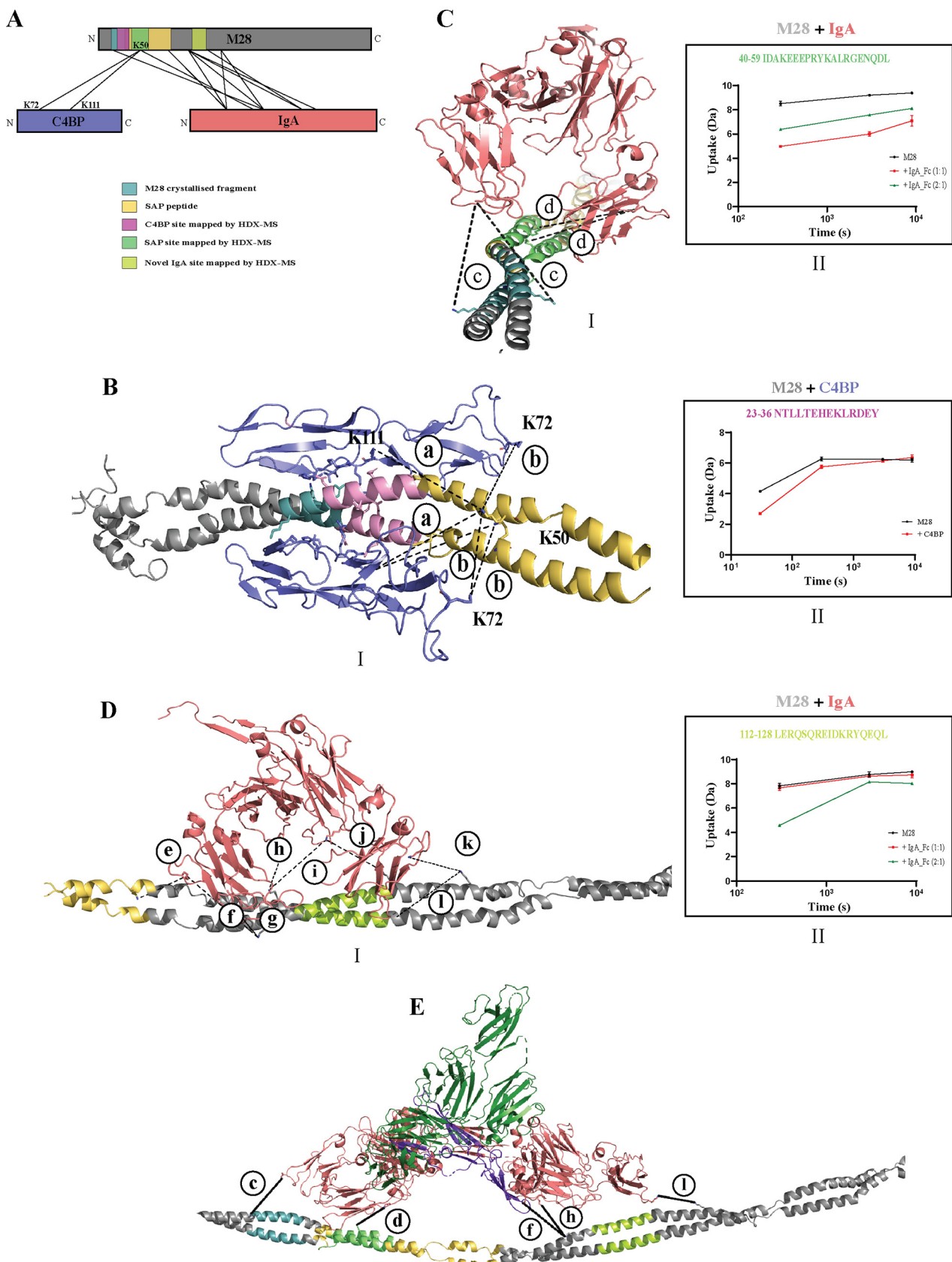

**FIG 5** Identified interaction interfaces of C4BP and IgA on M28. (A) Schematic depicting the binding regions of C4BP and IgA on M28 as identified by TX-MS and HDX-MS. The lines represent cross-links. The HDX-MS-mapped sites for C4BP are in pink, and the IgA sites are in light green and lemon green,

(Fig. S5), indicating that this is a low-affinity site, as suggested by the SPR data (Fig. 4A). The latter M28 site protected from deuterium uptake overlaps the novel IgA Fc interface identified by TX-MS (Fig. 5A and DII). Thus, both TX-MS and HDX-MS confirm the two distinct IgA-binding sites on M28. The two binding sites between IgA and M28 could result in the binding of either two single IgA Fc's (Fig. 5CI and Fig. 5DI) or one sIgA molecule, where a dimeric IgA is bridged by a J chain and a secretory component (Fig. 5E). Using a recently determined structure for sIgA (47) as the input for TX-MS, we showed that the binding between sIgA and M28 is supported by five unique interprotein cross-links (Fig. 5E). The binding of sIgA to two separate and possibly synergistic binding sites on M28 could explain why sIgA binding was more pronounced in the AP-MS experiments than C4BP binding, as described above (Fig. 4C).

Taken together, our AP-MS data in combination with the integrative structural mass spectrometry approach allowed us to propose two distinct models for the M28 interactions. In one model, two single IgA Fc monomers and a C4BP molecule would simultaneously bind to M28 (Fig. 6A), and in the other one, the IgA-binding sites would be occupied by sIgA alone (Fig. 6B). The results suggest that the domain arrangement of M28 is responsible for the formation of microenvironment-dependent protein interactions.

## DISCUSSION

The clinical manifestations of *S. pyogenes* are diverse (3). This bacterium presents itself on the skin and throat, causing localized infections, but can also breach the cellular layer to cause systemic infections. This forces *S. pyogenes* to adapt to different host microenvironments. In this study, we used a combination of MS-based methods to demonstrate that different streptococcal serotypes bind specifically to distinct sets of human proteins depending on the serotype and local microenvironment. These interactions were in turn mediated by one of the most abundant and widely studied surface-attached virulence factors, the M proteins. The established M-centered interaction networks recapitulated many of the previously identified M protein-human interactions and in addition highlight several so far functionally uncharacterized protein interactions. Interestingly, we note that the binding interactions were highly divergent between the analyzed *emm* types. A prominent example is the binding of fibrinogen to the A-C pattern and of C4BP to the E pattern M proteins. Fibrinogen is known to bind to the B repeats of the M protein (10, 15, 20), and C4BP is known to bind to the HVR domain (33–37), confirming that serotype-specific networks are highly dependent on the M protein domain arrangement. Therefore, our data suggest that once an interaction with fibrinogen has been established, an interaction with C4BP is not readily formed and vice versa. These results imply that sequence variability and the domain arrangement of M proteins dictate the type of human proteins that they can bind in a

**FIG 5** Legend (Continued)

respectively. (BI) Closeup view of the cross-linked site identified between M28 (gray helix) and C4BP (blue). The interaction interface on the crystallized M28 segment (PDB accession number 5HYP) is shown in cyan, and the SAP interacting with the IgA Fc domain is in yellow. Cross-links are observed between lysine residues K72 and K111 (numbered based on the full-length C4BP$\alpha$ chain) and K50 on our M28 construct. The cross-links are depicted as dotted lines, with the labels corresponding to a given spectrum in Fig. S4 and Table S2 in the supplemental material. Due to the dimeric nature of M28, several combinations of cross-links are possible. The HDX-MS-mapped binding site of C4BP on M28 is represented in pink. (II) Deuterium uptake graph for amino acids 23 to 36 on M28 alone (black) and on M28 and C4BP (red). This amino acid stretch is colored in pink in the M28 model in panel BI. (CI) Closeup view of the IgA Fc-binding interface on M28 identified by TX-MS. The cross-linked site overlapping the identified C4BP interaction interface and the previously identified M22-based IgA-binding SAP between the M28 (gray helix) SAP region (yellow) and the IgA Fc domain (red) is viewed down along the helix. The cross-links are depicted as dotted lines, with the labels corresponding to a given spectrum in Fig. S4 and Table S2. The HDX-MS-mapped site on M28 is denoted in light green. Black represents M28 alone, red is M28-IgA Fc (1:1 ratio), and green represents M28-IgA Fc (2:1). (II) Deuterium uptake graph for M28 and IgA. The region from aa 40 to 59 on M28 seems to bind IgA, the suggested high-affinity site. This site is marked in light green in panel CI. (DI) Closeup view of the novel interaction site between M28 (gray helix) and the IgA Fc domain (red). The cross-links are depicted as dotted lines, with the labels corresponding to a given spectrum in Fig. S4 and Table S2. The HDX-MS-identified IgA-binding site is denoted in lemon green. (II) Deuterium uptake graph for M28 and IgA. HDX-MS identified aa 112 to 128 (lemon green) on M28, the suggested low-affinity site. This is also denoted in panel DI in lemon green. Black represents M28 alone, red is M28-IgA Fc (1:1 ratio), and green represents M28-IgA Fc (2:1). (E) The possible secretory IgA Fc (red)-M28 model. Purple represents the J chain, and the secretory component is represented in dark green. The cross-links are depicted as dotted lines, with the labels corresponding to a given spectrum in Fig. S4 and Table S2. Light green and lemon green represent the HDX-MS-identified high-affinity and low-affinity sites of IgA Fc on M28. (The illustration in panel A was prepared in xVis.)

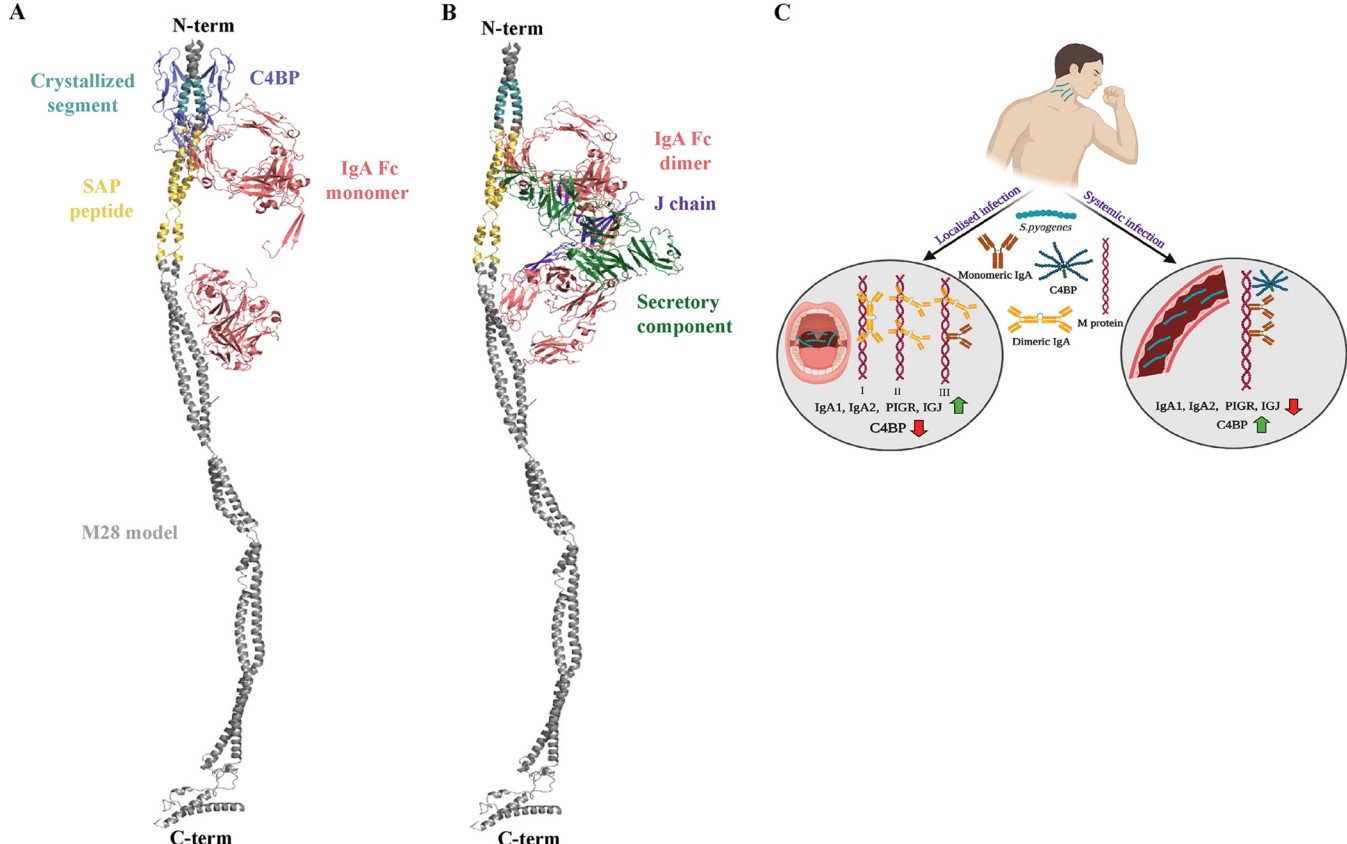

**FIG 6** Overview of IgA Fc and C4BP binding to M28. The homology model for the E-type M protein M28 is depicted as a gray helix. Cyan represents the X-ray-crystallized M28 domain, while yellow is the known IgA Fc-binding SAP peptide. (A) M28 model depicting the concomitant binding of C4BP (blue) and two IgA Fc monomers (red). (B) M28 model with secretory IgA Fc (red). Green represents the secretory component, and purple represents the J chain. (C) Schematic overview of M28 binding secretory IgA, monomeric IgA, and C4BP in different microenvironments in the cases of localized and systemic infections. (The illustration in panel C was created using BioRender.)

host microenvironment. The recruitment of prominent proteins in a particular niche can maximize the chances for successful immune evasion in that microenvironment.

Both the SA-MS and AP-MS data revealed a strong IgA interaction with M28, while no other serotype of M protein investigated in this study was observed to bind IgA to the same degree. The SAP derived from M22 (28) was previously shown to harbor an IgA-binding site. In our study, M28 is the only M protein that contains the SAP sequence. The novel secondary IgA-binding site, in contrast, is conserved between M28 and M89 (Fig. S1 and Fig. S5A). AP-MS analysis of M28 and M89 with human plasma and IgA from human serum reveals IgA binding to M89 as well. However, the binding of IgA to M89 is weaker than the binding to M28, and this could be due to the absence of SAP in M89, thus suggesting that the novel site is a weak-affinity IgA-binding site. IgA is the most abundant immunoglobulin on the mucosal surface. As *S. pyogenes* is known to localize at mucosal surfaces, strong binding of IgA to M28 could hence be warranted and likely plays a role in facilitating the bacteria to evade the first line of immune defense on the mucosal surface and facilitating bacterial adhesion to the mucosal cell surface. In fact, the M28 serotype has been reported to be one of the leading causes of puerperal sepsis (48–51). Persistent infections of the mucosal membrane by *S. pyogenes* can induce vascular leakage, thereby providing access of the bacterium to human plasma. Here, we tried to mimic localized infection conditions followed by systemic infection, and we observed that under such circumstances, the M28 interaction network gradually changes its composition from the predominant binding of secretory IgA in saliva to the binding of monomeric IgA and C4BP in plasma. This change is driven by the differences in protein concentrations in the host microenvironment. However, even at higher plasma concentrations (10% plasma),

secretory IgA is enriched to a greater extent with M28 than in the input sample, whereas C4BP is not enriched to the same extent under these conditions. Typically, bacterium-host relationships are well balanced. Sepsis is a relatively rare condition compared to uncomplicated local infections, implying that the evolution of bacterium-host relationships is predominately taking place in local host microenvironments and not in blood (26). In local microenvironments, secretory IgA is the major immunoglobulin. Our results support the following three models in a mucosal niche: (i) one dimeric IgA occupies both IgA-binding sites on M28, (ii) two dimers bind separately to the two sites, and finally, (iii) one dimer and one monomer could be engaged on M28 (Fig. 6C). However, the stoichiometry of IgA dimer binding under such conditions remains unexplored. During the course of an infection, there can be local damage to the mucosal membrane, causing leakage of the plasma exudate and thus creating an upsurge of C4BP in the local environment (34). Under such circumstances, the bacterium is known to encounter IgG from plasma, but *S. pyogenes* has many well-described virulence factors, like EndoS (52), SpeB (53), and IdeS (54), to circumvent IgG effects. This change in the local microenvironment may therefore drive binding to C4BP along with monomeric IgA (Fig. 6C). It has been reported that the binding of both IgA and C4BP to an M protein is crucial in inhibiting phagocytosis (32). C4BP is known to bind to the HVR of the M proteins, and IgA binds a semiconservative domain adjacent to the HVR site (32), which makes the concomitant binding of C4BP and monomeric IgA to M28 plausible, as previously suggested (28). Macrophages express IgA Fcα receptors; therefore, the binding of M proteins to IgA Fc could inhibit phagocytosis, and the binding of C4BP would prevent the activation of the complement pathway, thereby stopping the release of C5a and C3a and preventing anaphylaxis. As M proteins are known to be imperfectly coiled (55), there is a possibility that the binding of one protein at a certain site might introduce conformational changes in other parts of the coiled coil, thereby affecting the affinity of proteins on another site. In this case, the binding of monomeric IgA might induce a conformational change that promotes the binding of C4BP to the HVR of M28, in a manner similar to what has been shown for increased C4BP binding to the streptococcal surface mediated by IgG (56, 57). This could also explain the strong coupling seen between IgA and C4BP in the SA-MS and AP-MS experiments. We propose that M28 binds either secretory IgA or monomeric IgA and C4BP, depending on whether they cause a localized infection or a systemic infection (Fig. 6C). The structural model presented here is consistent with our finding of M28's dimeric IgA binding and concomitant binding to monomeric IgA and C4BP but in different ecological niches.

## MATERIALS AND METHODS

**Cloning, expression, and purification of recombinant proteins.** The M proteins were cloned, expressed, and purified at the Lund Protein Production Platform (LP3) (Lund, Sweden) and at the University of Oslo (Norway). The recombinant M proteins used in this study lacked the signal peptide and the LPXTG cell wall-anchoring motif. The open reading frames corresponding to the mature M proteins M1 (UniProt accession number Q99XV0; aa 42 to 448; gene *emm1*), M3 (UniProt accession number W0T370; aa 42 to 545; gene *emm3*), M5 (UniProt accession number P02977; aa 43 to 456; gene *emm5*), M28 (UniProt accession number W0T1Y4; aa 42 to 358; gene *emm28*), M49 (UniProt accession number P16947; aa 42 to 354; gene *emm49*), and M89 (UniProt accession number W0T3V8; aa 42 to 360; gene *emm89*) were cloned into a pET26b(+)-derived vector carrying a 6×His-HA-StrepII-TEV (six-histidine–hemagglutinin–Strep-tag II–tobacco etch virus protease recognition site) tag. The protein sequences are provided in Table S1 in the supplemental material. These proteins were expressed in *Escherichia coli* Tuner(DE3) induced with 1 mM isopropyl $\beta$-D-1-thiogalactopyranoside (IPTG) at an optical density at 600 nm ($OD_{600}$) of 0.6 at 18°C at 120 rpm (shaking) after 18 h. The cells were harvested by centrifugation at 8,000 × *g* at 4°C for 20 min, and the pellets were resuspended in buffer containing 50 mM $NaPO_4$, 300 mM NaCl, and 20 mM imidazole (pH 8) (buffer A) supplemented with EDTA-free complete protease inhibitor tablets (Roche). The cells were lysed using a French press at 18,000 lb/in². The cell lysate was cleared by ultracentrifugation at 244,000 × *g* (Ti50.2 rotor) for 60 min at 4°C, with subsequent passaging through a 0.45-$\mu$m syringe filter. The cell lysate was loaded onto a HisTrap HP column (GE Healthcare), followed by washing with 20 column volumes of buffer A, and the bound proteins were subsequently eluted with a 0 to 100% gradient of buffer B (50 mM $NaPO_4$, 300 mM NaCl, 500 mM imidazole [pH 8]). The fractions containing the protein of interest were pooled, dialyzed against 1× phosphate-buffered saline (PBS) (10 mM phosphate buffer, 2.7 mM KCl, 137 mM NaCl [pH 7.3]), and stored at −80°C until further use. The expression and purification of superfolder GFP (sfGFP) were described previously (17).

**Removal of the M28 affinity tag by TEV protease digestion.** For surface plasmon resonance (SPR), targeted cross-linking mass spectrometry (TX-MS), and hydrogen-deuterium mass spectrometry (HDX-MS),

M28 without the affinity tag was used. For the removal of the affinity tag, the M28 protein was treated with TEV protease at an enzyme-to-substrate mass ratio of 1:20. Dithiothreitol (DTT) was added to a final concentration of 1 mM, and the digestion mixture was transferred to a dialysis membrane (6,000- to 8,000-molecular-weight cutoff) and dialyzed against buffer A supplemented with 1 mM DTT at 16°C for 18 h. The mixture was passed through a HisTrap column (GE Healthcare) at room temperature (RT) with the same gradient and buffer as the ones described above. Fractions containing the cleaved M28 protein were collected and passed through a 0.2-$\mu$m syringe filter before loading onto a 26/600 Superdex 200-pg column (GE Healthcare) run with 1× PBS (pH 7.4) at 2.5 ml/min at 6°C. Fractions with TEV-cleaved, purified M28 were pooled and stored at −80°C until further use.

**Commercial proteins and human plasma and saliva.** Pooled human plasma (lot numbers 18944 and 27744) and pooled human saliva (catalog number IR100044P) were purchased from Innovative Research, USA. Pooled saliva was centrifuged at 1,500 × g for 15 min at 4°C followed by sterile filtration using 0.22-$\mu$m Steriflip filtration units (Millipore) and storage at −20°C until further use. IgA from human serum (lot number 0000085362) was purchased from Sigma-Aldrich, Germany. Purified human complement C4BP (catalog number A109, lot number 4a) was obtained from Complement Technology, USA. The recombinant human IgA Fc domain (catalog number PR00105) was purchased from Absolute Antibody, UK.

**Bacterial culture.** *S. pyogenes* serotype M1 (SF370) was obtained from the American Type Culture Collection (ATCC) (ATCC 700294), which was originally isolated from an infected wound. The other *S. pyogenes* isolates of serotypes M3, M5, M28, M49, and M89 used in this study were clinical isolates obtained from the blood of GAS-infected patients at Lund University Hospital and serotyped by the clinical microbiology department of the hospital. These bacteria were grown on blood agar plates, and single colonies were isolated and grown in Todd-Hewitt (TH) broth supplemented with 0.6% yeast extract at 37°C with 5% $CO_2$ for 16 h. Bacteria from the culture grown overnight were subcultured in TH broth with 0.6% yeast extract at 37°C with 5% $CO_2$ until the mid-logarithmic phase ($OD_{600}$ of 0.4 to 0.5). The cells were harvested by centrifugation at 3,500 × g for 5 min. The pellets were washed in HEPES buffer (50 mM HEPES, 150 mM NaCl [pH 7.5]) twice and recentrifuged at 3,500 × g for 5 min. The washed cells were resuspended in HEPES buffer to a 1% solution. These cells were further used for SA-MS experiments.

**Bacterial surface adsorption of human plasma proteins.** To capture human plasma proteins on the *S. pyogenes* surface, 400 $\mu$l of pooled normal human plasma was added to 100 $\mu$l of a 1% bacterial solution in six biological replicates for each strain. The samples were vortexed briefly and incubated in a shaker (500 rpm) at 37°C for 30 min. Cells were harvested by centrifugation at 5,000 × g for 5 min and washed three times with HEPES buffer, followed by centrifugations at 5,000 × g for 5 min, respectively. The cells were finally resuspended in 100 $\mu$l HEPES buffer. For limited proteolysis of surface-attached bacterial and human proteins, 2 $\mu$g of 0.5-$\mu$g/$\mu$l sequencing-grade trypsin (Promega) was added, and the digestion was allowed to proceed at 37°C with shaking at 500 rpm for 60 min. The reaction was stopped on ice, and the supernatant was collected by centrifugation at 1,000 × g for 15 min at 4°C. Any remaining bacteria in the supernatants were heat killed at 85°C in a shaker (500 rpm) for 5 min, prior to sample preparation for mass spectrometry.

**Affinity purification of human plasma, saliva proteins, and IgA.** For affinity purification (AP) reactions, 20 $\mu$g of recombinant affinity-tagged M proteins was incubated with 150 $\mu$l of a 50% Strep-Tactin Sepharose bead slurry (IBA) equilibrated in 1× PBS. Affinity-tagged sfGFP was used as a negative control in all experiments. Pooled normal human plasma (100 $\mu$l) or saliva (200 $\mu$l) or IgA (20 $\mu$g) was then incubated in a shaker (800 rpm) with the protein-bound beads at 37°C for 1 h. Every 1 ml of saliva was complemented with 10 $\mu$l of a protease inhibitor (Sigma). For saliva-plasma mixed-environment experiments, 100-$\mu$l saliva-plasma dilutions were made for 100% saliva, 1% plasma (99 $\mu$l saliva plus 1 $\mu$l plasma), 10% plasma (90 $\mu$l saliva plus 10 $\mu$l plasma), and 100% plasma and incubated with the protein-bound beads at 37°C with shaking at 800 rpm for 1 h. The beads were washed with 10 ml ice-cold 1× PBS (for plasma) and 4 ml ice-cold 1× PBS (for saliva, saliva-plasma dilutions, and IgA) at 4°C before eluting the proteins with 120 $\mu$l 5 mM biotin in 1× PBS at RT. To remove biotin from the eluted protein mixture, trichloroacetic acid (TCA) was added to a final concentration of 25%, and the mixture was incubated at −20°C for 16 h. The protein mixture was centrifuged at 18,213 × g for 30 min at 4°C. The pellets were washed two times in 500 $\mu$l and once in 200 $\mu$l ice-cold acetone by centrifugation at 18,213 × g for 10 min at 4°C. These pellets were then prepared for mass spectrometry.

**Cross-linking IgA Fc and C4BP with M28.** Ten micrograms of C4BP and 10 $\mu$g of IgA Fc were incubated separately with 10 $\mu$g of M28 in a final volume of 100 $\mu$l in 1× PBS for 30 min at 37°C with shaking at 800 rpm. To cross-link IgA Fc or C4BP to M28, a heavy/light disuccinimidylsuberate cross-linker (DSS-$H_{12}$/$D_{12}$; Creative Molecules Inc.) resuspended in 100% dimethylformamide (DMF) was added to final concentrations of 0, 100, 250, 500, 1,000, and 2,000 $\mu$M. The cross-linking mixture was then incubated at 37°C at 800 rpm (shaking) for 60 min. Before preparing the sample for MS analysis, the reaction was quenched by adding ammonium bicarbonate to a final concentration of 50 mM and incubating the mixture for 15 min at 37°C at 800 rpm (shaking).

**Sample preparation for mass spectrometry.** To denature the proteins, a solution containing 8 M urea–100 mM ammonium bicarbonate was added to the SA-MS, AP-MS, and cross-linked samples. The disulfide bonds were reduced with 5 mM Tris(2-carboxyethyl)phosphine hydrochloride (TCEP) at 37°C for 60 min and then alkylated with 10 mM iodoacetamide in the dark at room temperature for 30 min. The samples were diluted with 100 mM ammonium bicarbonate for a final urea concentration of <1.5 M, and 0.5 $\mu$g/$\mu$l sequencing-grade trypsin (Promega) was then added for protein digestion at 37°C for 18 h. Mass spectrometry samples for cross-linking reactions were prepared in a similar fashion as stated above, with an additional step of digestion with 0.5 $\mu$g/$\mu$l lysyl endopeptidase (Wako) at 37°C at

mSystems®

800 rpm (shaking) for 2 h after treatment with iodoacetamide, followed by dilution with ammonium bicarbonate and trypsin digestion. The digestion mixtures were quenched with 10% formic acid to a final pH of 2 to 3. The peptides were purified in a Sola$\mu$ horseradish peroxidase (HRP) 2-mg/1-ml 96-well plate (Thermo Scientific) according to the manufacturer's protocol. The eluted peptides were dried in a SpeedVac and resuspended in a solution containing 2% acetonitrile–0.1% formic acid with iRT peptides (58) (retention time peptides as an internal reference), followed by 5 min of sonication and brief centrifugation before mass spectrometry.

**Liquid chromatography-tandem mass spectrometry (LC-MS/MS).** The peptides were analyzed using data-dependent mass spectrometry analysis (DDA-MS) and data-independent mass spectrometry analysis (DIA-MS) on a Q Exactive HFX instrument (Thermo Scientific) connected to an Easy-nLC 1200 instrument (Thermo Scientific). The peptides were separated on an Easy-Spray column (50-cm column, column temperature of 45°C; Thermo Scientific) operated at a maximum pressure of $8 \times 10^7$ Pa. A linear gradient of 4% to 45% acetonitrile in aqueous 0.1% formic acid was run for 65 min for both DDA and DIA. For DDA, one full MS scan (resolution of 60,000 for a mass range of *m/z* 390 to 1,210) was followed by MS/MS scans (resolution of 15,000) of the 15 most abundant ion signals. The precursor ions with 2 *m/z* isolation width were isolated and fragmented using higher-energy collisional-induced dissociation (HCD) at a normalized collision energy of 30. The automatic gain controls were set as 3e6 for the full MS scan and 1e5 for MS/MS. For DIA, a full MS scan (resolution of 60,000 for a mass range of *m/z* 390 to 1,210) was followed by 32 MS/MS full fragmentation scans (resolution of 30,000) using an isolation window of 26 *m/z* (including a 0.5 *m/z* overlap between the previous and the next windows). The precursor ions within each isolation window were fragmented using higher-energy collisional-induced dissociation at a normalized collision energy of 30. The automatic gain controls were set to 3e6 for MS and 1e6 for MS/MS. The cross-linked peptides were analyzed by DDA. For DDA of cross-linked peptides, one full MS scan (resolution of 60,000 for a mass range of *m/z* 350 to 1,600) was followed by MS/MS scans (resolution of 15,000) of the 15 most abundant ion signals within an isolation width of 2 *m/z*.

**SA-MS and AP-MS data analysis.** MS raw data were converted to gzipped and Numpressed mzML (59) using the tool MSconvert from the ProteoWizard v3.0.5930 suite (60). All data were stored and managed using openBIS (61). SA-MS DDA-acquired spectra were analyzed using the search engine X! Tandem (2013.06.15.1-LabKey; Insilicos, ISB) (62), OMSSA (version 2.1.8) (63), and COMET (version 2014.02 rev.2) against an in-house-compiled database containing the *Homo sapiens* and *S. pyogenes* serotype M1 reference proteomes (UniProt proteome identifiers UP000005640 and UP000000750, respectively) complemented with common contaminants from other species, yielding a total of 22,155 protein entries and an equal number of reverse decoy sequences. AP-MS DDA data were analyzed using the same search engines as the ones described above, against an in-house-compiled database containing the *Homo sapiens* and *S. pyogenes* serotype M1 reference proteomes (UniProt proteome identifiers UP000005640 and UP000000750, respectively) complemented with all of the affinity-tagged M proteins and the sfGFP sequences as well as common contaminants from other species, yielding a total of 22,162 protein entries and an equal number of reverse decoy sequences. Fully tryptic digestion was used, allowing two missed cleavages. Carbamidomethylation (C) was set to static and oxidation (M) was set to variable modifications, respectively. The mass tolerance for precursor ions was set to 0.2 Da, and that for fragment ions was set to 0.02 Da. The identified peptides were processed and analyzed using the Trans-Proteomic Pipeline (TPP v4.7 Polar Vortex rev 0, build 201403121010) using PeptideProphet (64). The false discovery rate (FDR) was estimated with Mayu (version 1.07), and peptide spectrum matches (PSMs) were filtered with the protein FDR set to 1%, resulting in a peptide FDR of <1%.

The SA-MS and AP-MS DIA data were processed using the OpenSWATH pipeline (65). For DIA, spectral libraries from the above-described DDA data set were created in openBIS (61) using SpectraST (version 5.0, TPP v4.8.0 Philae, build 201506301157-exported [Ubuntu-x86_64]) in TPP (66). For DIA, raw data files were converted to mzXML using MSconvert and analyzed using OpenSWATH (version 2.0.1, revision c23217e). The retention time extraction window was $\pm$300 s, and *m/z* extraction was set at a 0.05-Da tolerance. Retention time was then calibrated using iRT peptides. Peptide precursors were identified by OpenSWATH (2.0.1), and PyProphet (2.0.1) was used to control false discovery rates of 1% at the peptide precursor level and 1% at the protein level. Next, TRIC (67) was used to align the runs in the retention time dimension and reduce the identification error by decreasing the number of missing values in the quantification matrix. Further missing values were requantified by TRIC (67). The resulting DIA data sets were analyzed using Jupyter Notebooks (version 3.1.1). For DIA, proteins identified by more than 3 peptides and enriched with a $\log_2$ fold enrichment of >1 (2-fold) with an adjusted *P* value of <0.05 using Student's *t* test were considered true interactors. However, for the saliva-plasma dilution DIA data, TRIC was not enabled. The intensities of the proteins were estimated by summing the intensities of the most intense three peptides for each protein relative to the total peptide intensities (without iRT) for that protein. The AP-MS data for commercial IgA with M28, M89, and GFP were analyzed in MaxQuant (1.6.10.43).

**Surface plasmon resonance analysis of M protein.** Binding experiments were performed on a Biacore X100 instrument (Cytiva Life Sciences, Uppsala, Sweden) with control software v.2.0. All the assays were carried out on a Sensor CM5 gold chip (Cytiva Life Sciences, Uppsala, Sweden) at 25°C. For the covalent immobilization of M1 and M28 molecules via amine groups on the gold surface, an amine coupling kit (Cytiva Life Sciences, Uppsala, Sweden) containing EDC [1-ethyl-3-(3-dimethylamino-propyl)carbodiimide] (75 mg/ml), NHS (*N*-hydroxysuccinimide) (11.5 mg/ml), and ethanolamine (1 M; pH 8.5) was used.

The CM5 chip was docked into the instrument, and the chip surface was activated according to the EDC/NHS protocol with PBS buffer as the running buffer before the immobilization procedure. The ligand (M1/M28) was injected for 7 min (flow rate of 10 $\mu$l/min) at a concentration of 0.01 mg/ml (in 10 mM acetate buffer, pH 5.0), followed by an injection of 1.0 M ethanolamine for 7 min (flow rate of 10 $\mu$l/min) in order to deactivate excess reactive groups. Once the targeted immobilization level (~2,500 response units [RU]) was

achieved, no further immobilization was carried out. Flow channel 2 (Fc_2) (active channel) was used for ligand immobilization, while flow channel 1 (reference channel) was used as a reference to investigate non-specific binding. Response units were recorded from the subtracted channel (flow channel 2 − flow channel 1), which was then used to evaluate the results of the analysis. For IgA as the analyte, concentration series including 0, 0.009375, 0.01875, 0.0375, 0.075, 0.15, and 0.3 $\mu$M were prepared. For C4BP as the analyte, concentration series of between 0 and 96 nM were prepared. The analytes were injected into the active (Fc_2) and reference (Fc_1) channels at the same time. Triplicate injections were done for each concentration series. The association time was set to 120 s, while the dissociation time was kept at 600 s. For the regeneration of the surface, 10 mM glycine-HCl (pH 2.5) was used at a flow rate of 10 $\mu$l/min.

**Evaluation of SPR analysis.** For the evaluation of the SPR analysis, the kinetic parameters were determined using Biacore Evaluation software (v.2.0) for binding analysis based on curve-fitting algorithms, which employs global fitting.

The data collected for each experiment were analyzed according to a 1-to-1 fitting model using the kinetic fitting programs that yield association rate ($k_a$), dissociation rate ($K_d$), and $K_D$ values and also by fitting the data to a heterogeneous binding model. Equilibrium binding analyses were performed by plotting the RU values measured in the plateau versus each concentration series.

First, binding was tested for the simplest 1-to-1 Langmuir binding model, which follows the equation

$$A + B \underset{K_d}{\overset{k_a}{\rightleftarrows}} AB \qquad (1)$$

where $A$ is the analyte, $B$ is the ligand, and $AB$ is the complex. The $k_a$ (M$^{-1}$ s$^{-1}$) is measured from the reaction in the forward direction, while the $K_d$ (s$^{-1}$) is measured from the reverse reaction.

Binding was also tested for the heterogeneous ligand model where the same analyte binds independently to multiple ligands or to several binding sites on the same ligand. The heterogeneous ligand model follows the equation

$$A + B_1 \underset{K_{d1}}{\overset{k_{a1}}{\rightleftharpoons}} AB_1$$

$$A + B_2 \underset{K_{d2}}{\overset{k_{a2}}{\rightleftharpoons}} AB_2 \qquad (2)$$

where $A$ represents the analyte; $B_1$ and $B_2$ represent two different ligands or two different binding sites on the same ligand, respectively; $AB_1$ and $AB_2$ represent the first and second complexes formed after the binding of the analyte to the surface; $k_{a1}$ and $k_{a2}$ are the association rates of the first and second complexes; and $k_{d1}$ and $k_{d2}$ represent the dissociation rates.

**TX-MS data analysis and computational modeling.** The UniProt accession numbers used for the *S. pyogenes* M28 protein and human C4BPa, C4BPb, IGHA1, and IGHA2 are W0T1Y4, P04003, P20851, P01876, and P01877, respectively. The tertiary structure of the M28 protein was characterized using the Rosetta comparative modeling (RosettaCM) protocol (45) from the Rosetta software suite (68) based on the previously generated full-length model of the M1 protein (27) as the homologue structure. For IgA and C4BP, PDB accession numbers 6LXW and 5HYP, respectively, were used. To analyze the interactions of M28 with IgA and C4BP, the TX-MS protocol was employed (18), through which computational docking models were generated and filtered out using distance constraints derived from MS-DDA data. A final round of high-resolution modeling was performed on the top selected models to repack the side chains using the RosettaDock protocol (69).

**HDX-MS sample preparation and data acquisition.** HDX-MS was performed in two separate runs on M28 with IgA Fc and M28 with C4BP. In each experimental run, HDX-MS was first performed on pure untagged M28 (1 mg/ml). Next, the following mixtures of M28 with the different ligands were prepared in PBS and subjected to HDX-MS: M28-IgA(Fc) at a 1:1 molar ratio, where each sample consisted of 1 $\mu$l of M28 (75 pmol/$\mu$l) mixed with 1 $\mu$l of PBS and 3 $\mu$l of IgA(Fc) at a concentration of 20 pmol/$\mu$l; M28-IgA(Fc) at a 2:1 molar ratio, where each sample consisted of 2 $\mu$l of M28 (75 pmol/$\mu$l) mixed with 3 $\mu$l of IgA(Fc) at a concentration of 20 pmol/$\mu$l; M28 (pure), a run where each sample consisted of 1 $\mu$l of M28 (75 pmol/$\mu$l) mixed with 4 $\mu$l of PBS; M28-C4BP, a run where each interaction sample consisted of 2 $\mu$l of M28 (75 pmol/$\mu$l) mixed with 5 $\mu$l of C4PB (1 to 2 pmol/$\mu$l); and M28 (pure), where each sample consisted of 2 $\mu$l of M28 (75 pmol/$\mu$l) mixed with 5 $\mu$l of PBS.

The HDX-MS analysis was performed using automated sample preparation on a Leap H/D-X Pal platform interfaced to an LC-MS system comprising an Ultimate 3000 micro-LC instrument coupled to an Orbitrap Q Exactive Plus MS instrument. Samples of M28 with and without ligand were diluted with 25 $\mu$l 10 mM PBS (pH 7.4) (for time zero samples) or with 25 $\mu$l HDX labeling buffer comprising deuterated PBS (dPBS) of the same composition prepared in D$_2$O, and the pH was adjusted to pH$_{(read)}$ 7.0 with DCl diluted in D$_2$O. The HDX reactions were carried out for 30, 300, and 3,000 s at 20°C. Labeling was quenched by dilution of the labeled sample with 30 $\mu$l of a solution containing 1% trifluoroacetic acid (TFA), 0.4 M TCEP, and 4 M urea at 1°C, and 50 $\mu$l of the quenched sample was directly injected and subjected to online pepsin digestion at 4°C on an in-house-packed (Poros AL 20-$\mu$m immobilized pepsin) 2.1- by 30-mm pepsin column. Online digestion and trapping were performed for 4 min using a flow rate of 50 $\mu$l/min with a running buffer of 0.1% formic acid (pH 2.5). The peptides generated by pepsin digestion were subjected to online solid-phase extraction (SPE) on a PepMap300 C$_{18}$ trap column (1 mm by 15 mm) and washed with 0.1% formic acid (FA) for 60 s. Thereafter, the trap column was switched in-line with a 1- by 50-mm Hypersil Gold reversed-phase analytical column with a particle size of

1.9 $\mu$m, and separation was performed at 1°C using a gradient of 5 to 50% mobile phase B over 8 min and then 50 to 90% mobile phase B for 5 min; the mobile phases were 0.1% formic acid (mobile phase A) and 95% acetonitrile with 0.1% formic acid (mobile phase B). Following separation, the trap and column were equilibrated at 5% organic content until the next injection. The needle port and sample loop were cleaned three times after each injection with a mobile phase consisting of 5% methanol (MeOH) and 0.1% FA, followed by 90% MeOH and 0.1% FA and a final wash of 5% MeOH and 0.1% FA. After each sample and blank injection, the pepsin column was washed by injecting 90 $\mu$l of a pepsin wash solution containing 1% FA–4 M urea–5% MeOH. In order to minimize carryover, a full blank was run between each sample injection. Separated peptides were analyzed on a Q Exactive Plus MS instrument equipped with a heated electrospray ionization (HESI) source operated at a capillary temperature of 250°C. For undeuterated samples ($t$ = 0 s), 1 injection was acquired using data-dependent MS/MS HCD for the identification of the generated peptides. For HDX analysis (all labeled samples and one at 0 s), MS full-scan spectra at settings of a 70,000 resolution, an automatic gain control of 3e6, a maximum injection time (IT) of 200 ms, and a scan range of 300 to 2,000 Da were collected.

**HDX-MS data analysis.** PEAKS Studio X (Bioinformatics Solutions Inc., Waterloo, Canada) was used for peptide identification after pepsin digestion of undeuterated samples (i.e., 0-s time point). The search was done on a FASTA file comprising only the sequences of the analyzed proteins, and search criteria were a mass error tolerance of 15 ppm and a fragment mass error tolerance of 0.05 Da, allowing fully unspecific cleavage by pepsin.

Peptides identified by PEAKS with a peptide score value of a log $P$ value of >25 and no modifications were used to generate peptide lists containing the peptide sequence, charge state, and retention time for the HDX analysis. HDX data analysis and visualization were performed using HDExaminer version 3.01 (Sierra Analytics Inc., Modesto, CA, USA). Due to the comparative nature of the measurements, the deuterium incorporation levels for the peptic peptides were derived from the observed mass differences between the deuterated and nondeuterated peptides without back-exchange correction using a fully deuterated sample. HDX data were normalized to 100% $D_2O$ content with an estimated average deuterium recovery of 75%. Peptide deuteration was determined from the average of all high- and medium-confidence results, with the first two residues of each peptide set to be unable to retain deuteration. The allowed retention time window was set to ±0.5 min. Heat map settings were uncolored proline and heavy smoothing, and the difference heat maps were drawn using the residual plot as a significance criterion (±1 Da). The spectra for all time points were manually inspected; low-scoring peptides, e.g., obvious outliers, and peptides where retention time correction could not be made consistent were removed.

## SUPPLEMENTAL MATERIAL

Supplemental material is available online only.

**FIG S1**, EPS file, 2.3 MB.
**FIG S2**, EPS file, 2.6 MB.
**FIG S3**, PDF file, 0.6 MB.
**FIG S4**, PDF file, 0.4 MB.
**FIG S5**, EPS file, 1.8 MB.
**TABLE S1**, DOCX file, 0.1 MB.
**TABLE S2**, DOCX file, 0.1 MB.
**TABLE S3**, XLSX file, 0.1 MB.

## ACKNOWLEDGMENTS

Gene cloning, protein expression, and purification of the M protein and sfGFP were performed at the Lund Protein Production Platform (LP3), Lund University, Sweden (www.lu.se/lp3). We highly acknowledge Daniel Hatlem from D.L.'s lab for his guidance during recombinant protein purification. We thank Oonagh Shannon for access to the different *S. pyogenes* serotypes. We gratefully acknowledge support from the Swedish National Infrastructure for Biological Mass Spectrometry (BioMS).

This research was supported by the Viral and Bacterial Adhesin Network Training (ViBrANT) Program funded by the European Union's Horizon 2020 Research and Innovation Program under Marie Sklodowska-Curie grant agreement number 765042 to J.M. and D.L.; the Swedish Research Council (2019-01646) to J.M.; the Foundation of Knut and Alice Wallenberg (2016.0023 and 2019.0353) to J.M. and L.M.; and the Österlunds Stiftelse to J.M. and R.L. H.K. was supported by the Swiss National Science Foundation (early postdoc mobility grant number P2ZHP3_191289).

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
