## [Reviewer comments · mSystems]

Streptococcus pyogenes forms serotype and local environment-dependent inter-species protein complexes

Sounak Chowdhury, Hamed Khakzad, Gizem Bergdahl, Rolf Lood, Simon Ekstrom, Dirk Linke, Lars Malmström, Lotta Happonen, and Johan Malmström

Corresponding Author(s): Johan Malmström, Lund University

Review Timeline:

Submission Date:	March 5, 2021
Editorial Decision:	June 1, 2021
Revision Received:	July 30, 2021
Editorial Decision:	August 8, 2021
Revision Received:	August 24, 2021
Accepted:	September 6, 2021

Editor: Frank Schmidt

Reviewer(s): The reviewers have opted to remain anonymous.

Transaction Report:

DOI: <https://doi.org/10.1128/mSystems.00271-21>

May 19, 2021

Dr. Johan Malmström
Lund University
Lund
Sweden

Re: mSystems00271-21 (Streptococcus pyogenes forms serotype and local environment-dependent inter-species protein complexes)

Dear Dr. Johan Malmström:

Thank you for submitting your manuscript to mSystems. We have completed our review and I am pleased to inform you that, in principle, we expect to accept it for publication in mSystems. However, acceptance will not be final until you have adequately addressed the reviewer comments.

Thank you for the privilege of reviewing your work. Below you will find instructions from the mSystemseitorial office and comments generated during the review.

Preparing Revision Guidelines

For complete guidelines on revision requirements, please see the Instructions to Authors at <https://msystems.asm.org/sites/default/files/additional-assets/mSys-ITA.pdf>. **Submissions of a paper that does not conform to mSystems guidelines will delay acceptance of your manuscript.**

Sincerely,

Frank Schmidt

Editor, mSystems

Journals Department
Reviewer comments:

Reviewer #1 (Comments for the Author):

Chowdhury et al. report mass spectrometric analysis of human plasma proteins bound by six different M type strains of group A Streptococcus. They also identified human plasma proteins bound by purified forms of the six M proteins. The authors focused on one particular M protein type, M28, and carried out SPR analysis of its interactions with IgA and C4BP, as well as analysis of M28 protein interactions with human proteins contained in varying proportions of human saliva and plasma. Lastly, the authors carried out crosslinking coupled with mass spectrometry and hydrogen-deuterium exchange experiments to map the IgA- and C4BP-binding sites on M28 protein. From this, they identified a novel IgA-binding site on the M28 protein.

Overall, this work adds to the knowledge about interactions of various M protein types with human proteins, but an opportunity to characterize novel protein interactors appears to have been missed, as the authors turned their attention to interactions that have previously been characterized (some in quite detail). The manuscript appears to be written for mass spectrometry experts rather than a more general audience, and could use more thorough explanation of technical points and analysis. This made it difficult to gauge the validity of some of the points (see below in Major Issues).

The most problematic issue was one of overstatement regarding the import of the findings. The authors state (L. 616), "The ability to alter the protein interaction network depending on the host microenvironment allows *S* [missing period] pyogenes to initiate critical immune evasion strategies in different ecological niches of relevance for both mucosal and systemic infections." However, there is no evidence that *S. pyogenes* alters the protein interaction network. *S. pyogenes* seems to bind whatever is available. In saliva, this is (dimeric) sIgA and in plasma, it is (monomeric) IgA and C4BP. Puzzlingly, the authors find less C4BP is bound in 1% and 10% plasma than expected based on affinity measurements. There is no mechanistic explanation for this. The authors seem to imply a causal relationship by stating (L. 568), "Collectively, these results show that M28 binds secretory IgA in saliva and C4BP binding on M28 only becomes accentuated in the absence of secretory IgA." They further state (L. 639), "The results imply that sequence variability and the domain arrangement of M proteins can result in affinity differences to facilitate recruitment of human

proteins that maximizes the chances for successful immune evasion in different microenvironments." There is certainly a negative correlation between sIgA and C4BP binding but no evidence for causality, as there appears to be no mechanistic explanation for the decreased C4BP binding in the presence of sIgA. It appears that IgA and sIgA bind M28 protein in the same general manner, with the SAP and the novel site on M28 protein being occupied by two molecules of IgA or by a single dimeric molecule of sIgA. So, it appears not to be the case that sIgA blocks C4BP binding, and the reason for the lower than expected C4BP binding in <100% plasma remains obscure. In addition, why binding less C4BP in a mucosal environment would be advantageous to *S. pyogenes* is not evident and so it can hardly be called a strategy, and there is no evidence that the differences in recruited proteins in mucosa (sIgA) vs plasma (IgA and C4BP) result in maximizing immune evasion.

Other Major Issues

1. As mentioned above, a novel IgA-binding site on M28 protein was identified by cross-linking and mass spectrometry. Since this is a novel site, additional means of validation should be provided (e.g., demonstration that this site is sufficient, in the absence of the SAP sequence, for binding IgA).
2. A large number of proteins are presented in Figure 2A. Presumably, some novel interactions have been detected. A table listing the significant interactors with annotation of the proteins should be provided. This would likely be very helpful to the community. An explanation of the Z-score should be provided. What is being measured for the Z-score? And one of the categories is "Cell adhesion and cytoskeleton proteins." Were any cytoskeletal proteins found, and if so, what is the explanation for this?
3. Figure 2B is confusing. What does it mean when a protein is not within one of the M protein ellipses? Is it then common to all the M proteins depicted? What does the thickness of the lines imply? There are line colors that don't belong to the red-blue gradation, i.e., yellow ones (e.g., connecting FIBB and PON1). What do these mean? What does it mean when two or more proteins are connected by anticorrelation (red) lines? For example, PROP, ALBU, CFAB appear to be connected by red line. Presumably, this means that when CFAB is present, then ALBU is not. But then what happens to PROP?
4. What are the values on the y-axis on Figure 2C?
5. In Figure 3A, presumably, these data reflect the log₂ differences between the pull-downs for M1 protein vs. GFP from serum. So, why is there a dot for GFP on the volcano plot? Was GFP added to plasma? This question applies to Supplementary Figure 2.
6. The fits to the SPR data, even with a heterogeneous model, do not appear to be very good. Is there a statistical measure of the goodness-of-fit (e.g., χ^2)? And since additional KD's increase the number of parameters, it is important to have a means of cross-validation (i.e., exclusion of some data points from the fit, such that the fit is independent of these points; and evaluation of the errors of these points to the independent fit).
7. Figure 4C. Relative intensity is reported, but relative to what? It is difficult to gauge binding from the presentation, as the relative intensities for input and pulldown are different. Perhaps, the ratio of pulldown to input would be more meaningful.
8. The M1 strain seems not to bind fibrinogen or very little fibrinogen, while the M1 protein is competent for binding fibrinogen. This requires some comment.
9. Factor H has been noted to interact with M5 protein, but was not detected in this study. Was factor H present? The authors should comment on this.

Minor Issues

1. L. 91, "Binding of M proteins to IgA-Fc is believed to block the binding of IgA to CD89 ..." There is no room for "belief" in science. Either there is evidence or not.
2. L. 96, "inhibiting the classical complement pathway." C4BP also inhibits the lectin pathway.

3. Fig. 2A: One set of M3 data is in the M28 data group.
4. Fig. 2 Legend and elsewhere: "Person correlation network analysis" Presumably "Pearson" is meant here.
5. Fig. 2. Legend: "Each sphere represents a protein cluster." There are no spheres, only circles and ellipses.
6. L. 512, "no structural model for any E type M interspecies protein complex, we selected M28 for further structural characterization with a particular focus on the binding with IgA and C4BP as outlined in Figure 1B." This is inaccurate as there is an X-ray crystal structure of M28 protein bound to C4BP.
7. L. 519, "We immobilized the M proteins (ligand) on the sensor chip and injected IgA (analyte) over them to mimic the M proteins protruding out from the bacterial surface and the immunoglobulins floating in the plasma." This is an overstatement, as the M proteins were attached through random amine coupling.
8. The authors should note that the interaction between M28 protein and C4BP is likely to be characterized by avidity, as there are 7 α chains for C4BP and M28 protein is a dimer.
9. L. 549, "These AP-MS experiments were performed using 100% saliva, 1% plasma, 10% plasma in saliva and 100% plasma to mimic conditions during a local infection followed by a systemic infection." This is confusing. It makes it appear that the 1% plasma sample had no saliva.
10. Figure 3C. "SF370" is used instead of "M1" and the additional labeling of M28 and M89 require explanation.
11. L. 638, "A notion that requires further investigations." This sentence is ungrammatical; it lacks a verb.
12. In the Methods, in some places, a space between numbers and units is missing and, in some places, degree signs are missing.
13. L. 130, Values in rpm depend on the rotor. Instead, rcf values should be reported.
14. L. 190, "For affinity purification (AP) reactions 20 μ g of recombinant affinity-tagged M proteins was charged on Strep-Tactin Sepharose beads (IBA) equilibrated in 1x PBS." What does "charged" mean?

Reviewer #3 (Comments for the Author):

The elegant study by Chowdhury et al. presents the interaction networks of group A streptococcal M-proteins in plasma (and saliva). The authors show that the interactions depend on (i) GAS serotype and (ii) particularly M protein domain arrangement. Divers range of network analyses showed that plasma proteins albumin, IgG1, IgG4, and IgA2 are associated with all tested M proteins. In contrast, C4BP α , IgA1, alpha-1-antitrypsin (A1AT), and PROS were mainly associated with M28 and IgA2 showed a significant enrichment with M28. Since the interaction networks of M28 showed such a unique pattern and emm pattern E stains are not well characterized, the authors focused in the second half of the manuscript mainly on M28-interactions. Major conclusions of the study are: (i) M28 binds secretory/polymeric IgA (from saliva) and due to a potential vascular leakage, this binding can be replaced by (ii) monomeric IgA and/or C4BP from plasma. Based on these results the authors conclude that GAS adapt to specific microenvironments by forming protein complexes with host proteins, which in turn helps them to avoid immune detection at the border of mucosal and vascular compartments.

Overall, the study is well designed and performed, clearly presented, and of high clinical relevance. Although the authors used mainly MS-based approaches, major findings, e.g., M28-C4BP and M28-IgA interactions, are also validated and even quantified (calculated K_d-values). I have just a few minor comments.

Minor comments:

Lines 60-67: To my knowledge, the classification of emm patterns A-E is not solely based on M protein sequences (domain arrangement). This classification also includes the chromosomal arrangement of emm and emm-like genes (such as enn and mrp). Emm patterns B and C are rare and grouped together with pattern A strains, due to their structural similarities (i.e., all have an SF-1 emm gene and lack mrp). Pattern D and E strains have mrp and the SF-3 form of enn, but are distinct in that their emm genes are of the SF-1 and SF-2 forms, respectively.

Figure 4A. It would be nice if the KD values would be included within the respective graph.

Figures S3 (and 3A)/lines 521-533: The authors tested two fitting models and conclude that heterogeneous ligand model gave the best fit for M28-IgA and M1-IgA interactions, while C4BP-M28 interaction showed a better fitting in a 1-1 model. Just by looking at the Figure S3, I would totally agree with such conclusion concerning M28-IgA interaction. However, I do not see a difference between the fitting models in case of Fig. S3B and C. Could the authors please provide the Chi2 values (measure of the accuracy of the fitting) within the respective graphs, which were probably used to judge which fitting was the best.

As the authors state in the introduction, binding of M proteins to IgA is believed to prevent IgA-effector functions and inhibits phagocytosis thereby promoting bacterial virulence. However, (i) not every throat infection results in severe infections, (ii) emm28 strains account for a smaller proportion of severe infections (usually among the top 5), and (iii) secretory IgA is rather a weak opsonin in general. Could the authors please discuss how the identified binding sites (M28 and IgA/C4BP) and the resulting binding would impair the function of C4BP and IgA and therefore contribute to the virulence of GAS.

Below we provide the point-by-point responses to the reviewers. For your easy understanding the reviewer comments have been numbered and our response to the reviewer's comments are in italics.

Reviewer #1 (Comments for the Author):

Chowdhury et al. report mass spectrometric analysis of human plasma proteins bound by six different M type strains of group A Streptococcus. They also identified human plasma proteins bound by purified forms of the six M proteins. The authors focused on one particular M protein type, M28, and carried out SPR analysis of its interactions with IgA and C4BP, as well as analysis of M28 protein interactions with human proteins contained in varying proportions of human saliva and plasma. Lastly, the authors carried out crosslinking coupled with mass spectrometry and hydrogen-deuterium exchange experiments to map the IgA- and C4BP-binding sites on M28 protein. From this, they identified a novel IgA-binding site on the M28 protein.

Overall, this work adds to the knowledge about interactions of various M protein types with human proteins, but an opportunity to characterize novel protein interactors appears to have been missed, as the authors turned their attention to interactions that have previously been characterized (some in quite detail). The manuscript appears to be written for mass spectrometry experts rather than a more general audience, and could use more thorough explanation of technical points and analysis. This made it difficult to gauge the validity of some of the points (see below in Major Issues).

We agree that the manuscript was partly written for mass spectrometry experts. We have added more thorough technical explanations to the manuscript to make it more accessible for a wider research community. For easy understanding changes have been made in L.443, L. 750 and L.771. The structural mass spectrometry section (L.575-588) has been re-formatted and a brief summary of the methods used have been presented.

The most problematic issue was one of overstatement regarding the import of the findings. The authors state (L. 616), "The ability to alter the protein interaction network depending on the host microenvironment allows S [missing period] pyogenes to initiate critical immune evasion strategies in different ecological niches of relevance for both mucosal and systemic infections." However, there is no evidence that S. pyogenes alters the protein interaction network. S. pyogenes seems to bind whatever is available. In saliva, this is (dimeric) sIgA and in plasma, it is (monomeric) IgA and C4BP. Puzzlingly, the authors find less C4BP is bound in 1% and 10% plasma than expected based on affinity measurements. There is no mechanistic explanation for this. The authors seem to imply a causal relationship by stating (L. 568), "Collectively, these results show that M28 binds secretory IgA in saliva and C4BP binding on M28 only becomes accentuated in the absence of secretory IgA." They further state (L. 639), "The results imply that sequence variability and the domain arrangement of M proteins can result in affinity differences to facilitate recruitment of human proteins that maximizes the chances for successful immune evasion in different microenvironments." There is certainly a negative correlation between sIgA and C4BP binding but no evidence for causality, as there appears to be no mechanistic explanation for the decreased C4BP binding in the presence of sIgA. It appears that IgA and sIgA bind M28 protein in the same general manner, with the SAP and the novel site on M28 protein being occupied by two molecules of IgA or by a single dimeric molecule of sIgA. So, it appears not to be the case that sIgA blocks

C4BP binding, and the reason for the lower than expected C4BP binding in <100% plasma remains obscure. In addition, why binding less C4BP in a mucosal environment would be advantageous to *S. pyogenes* is not evident and so it can hardly be called a strategy, and there is no evidence that the differences in recruited proteins in mucosa (sIgA) vs plasma (IgA and C4BP) result in maximizing immune evasion.

*We have carefully considered the reviewers alternative explanation and agree that some parts of the manuscript were not well formulated giving the erroneous impression that *S. pyogenes* is actively altering the protein network depending on local microenvironment. We have reformatted the statement on L625 and we now state that “. The results suggest that the domain arrangement of M28 is responsible for the formation of microenvironment dependent protein interactions.” In addition to it we modified lines L.539, L.550, L.109, L110.*

As pointed out by the reviewer, it is puzzling that the concentration of C4BP bound to M28 in 1% and 10% saliva is lower than expected. To make sure that this observation is consistent, we repeated the experiment and find highly similar results (see Figures below (I)-reported in the manuscript and (II)-repeated experiment). This makes it quite clear that the concentration of C4BP bound to M28 is consistently lower than expected in the mixed saliva and plasma samples and that there is a negative correlation between sIgA and C4BP. However, as we have no mechanistic explanation for this observation we have down-toned the statements on (L. 571) to “Collectively, these results show that M28 binds secretory IgA in saliva and in plasma and binds IgA and C4BP in plasma.” and (L. 646) to “The results imply that sequence variability and the domain arrangement of M proteins dictates the type of human proteins it can bind in a host micro-environment. Recruitment of prominent proteins in a particular niche can maximize the chances for successful immune evasion in that microenvironment.”

Other Major Issues

1. As mentioned above, a novel IgA-binding site on M28 protein was identified by cross-linking and mass spectrometry. Since this is a novel site, additional means of validation should be provided (e.g., demonstration that this site is sufficient, in the absence of the SAP sequence, for binding IgA).

We thank the reviewer for this suggestion. First, we would like to point out that both the TX-MS and the HDX-MS analysis suggest that IgA interacts with a secondary site on the M28 (aa-83-150 identified by TX-MS and aa-112-128 identified by HDX-MS). To make this clearer we have changed the text on L.575-626 and made a new version of Figure 5.

To determine if this secondary binding site is sufficient for IgA binding in the absence of the SAP sequence, we performed two additional experiments. In these experiments, we used M89 which does not have a SAP sequence but a highly conserved secondary IgA binding site compared to M28 as shown in the sequence alignment in the new supplementary figure (Figure S5-1). The purple colour denotes SAP sequence which seems to be present only in M28 and the red denotes the novel IgA binding site that we report in the manuscript; which seems to be conserved between M28 and M89. To understand whether the secondary site contributes to IgA binding we re-analysed our AP-MS data with human plasma for GFP, M28 and M89 (Fig. S5-B). IgA is statistically enriched on M89 compared to GFP however the level of binding is lower as compared to M28. These results show that the secondary site is sufficient to bind IgA but likely with a much lower affinity. The presence of both SAP and secondary site on M28 leads to higher levels of IgA binding. To further prove this, we performed AP-MS with commercial IgA from human serum. AP-MS was performed individually with 20 µg of IgA and 20 µg of GFP, M28 and M89. We again observe that M89 enrich IgA however the levels are lower than M28 due to the absence of SAP sequence (Fig. S5-C). We have added a new supplement figure (Fig. S5) with these results and added this finding to the result section on L.609 and the discussion section L.656.

2. A large number of proteins are presented in Figure 2A. Presumably, some novel interactions have been detected. A table listing the significant interactors with annotation of the proteins should be provided. This would likely be very helpful to the community. An explanation of the Z-score should be provided. What is being measured for the Z-score? And one of the categories is "Cell adhesion and cytoskeleton proteins." Were any cytoskeletal proteins found, and if so, what is the explanation for this?

We apologize for not adding this information in the first version of the manuscript. We have added a new supplementary table (ST3) of the identified 92 proteins with their uniprot ID's, their protein family group, the number of peptides and the MS measured protein intensities across all samples.

We have added a new sentence in L. 443 explaining the principles of the Z-score "The clustering of the strains in the heatmap is based on z-score; where the z-score measures the standard deviation of a protein intensity from the mean intensity of that protein across all strains". We also added a sentence to the Figure legend of 2A L. 750 and 3B L. 771 - "Z-score value of -4 to +4 represents the standard deviation from the mean protein intensity (0). Red represents highly enriched proteins while blue represents less abundant proteins."

*We do identify cell adhesion proteins such as Integrin beta-3 (ITB3), Integrin alpha-IIb and Ankyrin-1. In depth MS analysis of plasma proteomes typically uncovers many tissue- and cell-specific proteins, among them cytoskeletal and cell adhesion proteins, which are likely remnants from cells and tissues (for more information see Malmström et al Nature communications 2016). These proteins are typically found in minute amounts in plasma but some of them specifically becomes enriched on the *S. pyogenes* surface indicating that these*

proteins bind to the bacterial surface. But further investigations are required to clarify if these interactions are specific.

3. Figure 2B is confusing. What does it mean when a protein is not within one of the M protein ellipses? Is it then common to all the M proteins depicted? What does the thickness of the lines imply? There are line colors that don't belong to the red-blue gradation, i.e., yellow ones (e.g., connecting FIBB and PON1). What do these mean? What does it mean when two or more proteins are connected by anticorrelation (red) lines? For example, PROP, ALBU, CFAB appear to be connected by red line. Presumably, this means that when CFAB is present, then ALBU is not. But then what happens to PROP?

Thanks for pointing this out. *We have added the missing information to the text on L.456 and Figure 2B. (L. 752) “Pearson correlation network analysis of human plasma proteins across six different serotypes. Colored ellipses are drawn around each M serotype to represent protein groups enriched on the individual serotypes. Each protein is depicted by a circle and colored according to the protein family they belong to. Proteins present in the ellipse bind strongly to particular serotypes. The ellipses of M3-M5 and M49-M89 are overlapping as they bind to common proteins while M1 and M28 have non-overlapping ellipse. Proteins outside the ellipse bound to most of the strains. Blue lines represent strongly correlating proteins ($r^2 > 0.9$), while red line lines represent mutual exclusive proteins or negatively correlating proteins ($r^2 < -0.6$). Thickness of the line represents how strongly the proteins are correlated”*

4. What are the values on the y-axis on Figure 2C?

This information went missing in the figure legend. We have added a sentence to figure legend of Figure 2C (L. 763) – “The x and y-axis represent the protein intensities”.

5. In Figure 3A, presumably, these data reflect the log₂ differences between the pull-downs for M1 protein vs. GFP from serum. So, why is there a dot for GFP on the volcano plot? Was GFP added to plasma? This question applies to Supplementary Figure 2.

It is correct that the volcano plot in Figure 3A and Supplementary Figure 2 reflect the log₂ difference of human proteins enriched on M proteins and GFP. GFP and the M proteins are the baits used to trap plasma proteins and are released from the affinity matrix in the final elution step and identified along with the other interactors by MS. For clarity the bait proteins are typically removed from the volcano plots, but we missed to do so this time. We have now removed the bait proteins from the volcano plots and mentioned this in the legend of Figure 3. (L.770) & Supplementary Figure 2 (L. 850)

6. The fits to the SPR data, even with a heterogeneous model, do not appear to be very good. Is there a statistical measure of the goodness-of-fit (e.g., χ^2)? And since additional KD's increase the number of parameters, it is important to have a means of cross-validation (i.e., exclusion of some data points from the fit, such that the fit is independent of these points; and evaluation of the errors of these points to the independent fit).

We have added a new table with the χ^2 values of each interactions and the fitted models in the Supplementary Figure 3 (S3). As you see in the Table (S3-D), for M28-IgA and M1-IgA interactions, the χ^2 values of heterogeneous binding models are lower than the ones for 1-1 binding models. On the other hand, for M28-C4BP binding, 1-1 binding model gives the lowest

chi² value. These values are consistent with our findings and statistically support the interaction models that we propose in our manuscript.

7. Figure 4C. Relative intensity is reported, but relative to what? It is difficult to gauge binding from the presentation, as the relative intensities for input and pulldown are different. Perhaps, the ratio of pulldown to input would be more meaningful.

Thank you for the comment. We have made a new version of Figure 4C. In this new version of the Figure, the input and pulldowns are similarly scaled for each protein and plotted in the same graph. As mentioned above, this experiment was repeated during revision with the same results. In this new Figure, protein intensities reflect the ratio of the individual intensity to the mean of the highest protein intensity across all the dilutions expressed as percentage. For proper understanding, we have modified the Figure 4C legend (L.791) to “To understand the levels of proteins enriched on M28 in different dilutions the data is represented as a comparison between input and the AP-MS. Relative intensity of peptides (Y-axis) plotted against 0, 1, 10 and 100% plasma concentration (X-axis) mimicking vascular leakage for IGHA1 (I), IGHA2 (II), PIGR (III), IGJ (IV), C4BPA (V) and IGHG1 (VI). Relative protein intensity (input or pulldown) reflects the ratio of the individual intensity to the mean of the highest protein intensity across all the dilution expressed as percentage. Green represents input samples and purple represents AP-MS with M28.”

8. The M1 strain seems not to bind fibrinogen or very little fibrinogen, while the M1 protein is competent for binding fibrinogen. This requires some comment.

In our surface adsorption experiments, we identify both human and bacterial proteins. These experiments show that the copy per cell numbers of the M-proteins vary. In particular, SF370 has for unknown reason a much lower copy number of the M protein compared to for example AP-1 (another M1 strain not included in this study) and the other strains analysed in this manuscript. We think this is the reason for the relatively low amount of fibrinogen found on the SF370 surface. When we use the same amount of M protein in the AP-MS experiments (20 µg for all M proteins), the levels of fibrinogen were high, supporting this notion. We have added a comment to the manuscript on L. 490.

9. Factor H has been noted to interact with M5 protein, but was not detected in this study. Was factor H present? The authors should comment on this.

We do confirm that Factor H binds to the M5 strain in our study. However, we used the less informative Uniprot ID in Figure 2 making this observation easy to overlook. The protein interactors are now annotated with the protein names and provided as a separate supplementary table (ST. 3).

Minor Issues

1. L. 91, "Binding of M proteins to IgA-Fc is believed to block the binding of IgA to CD89 ..." There is no room for "belief" in science. Either there is evidence or not.

We have modified the sentence (L. 91) to “Binding of M proteins to IgA-Fc blocks the binding of IgA to CD89, thus preventing IgA-effector functions, inhibiting phagocytosis and promoting bacterial virulence”.

2. L. 96, "inhibiting the classical complement pathway." C4BP also inhibits the lectin pathway.

We changed the sentence on L. 96 to: "C4BP bound to the M protein sequesters C4b from plasma, and acts as a co-factor for the degradation of C4b by complement factor P^{33,37,38}, thereby inhibiting the complement pathway and phagocytosis of the bacterium"

3. Fig. 2A: One set of M3 data is in the M28 data group.

Thank you, for noticing it. One of the M3 replicate is an outlier thereby clustering in the M28 data group. We have added a new sentence to the Figure 2A legend (L. 749) stating that "One of the M3 replicate is an outlier thereby falling in the X clade".

4. Fig. 2 Legend and elsewhere: "Person correlation network analysis" Presumably "Pearson" is meant here.

Yes, we meant pearson in the text. We have corrected this in the Figure 2B and 3C legend (L. 752 and L. 773).

5. Fig. 2. Legend: "Each sphere represents a protein cluster." There are no spheres, only circles and ellipses.

We have reformatted the Figure legend to make it clear to the readers (L. 752-761). We now state that "Pearson correlation network analysis of human plasma proteins across six different serotypes. Colored ellipses are drawn around each M serotype to represent protein groups enriched on the individual serotypes. Each protein is depicted by a circle and colored according to the protein family they belong to. Proteins present in the ellipse bind strongly to particular serotypes. The ellipses of M3-M5 and M49-M89 are overlapping as they bind to common proteins while M1 and M28 have non-overlapping ellipse. Proteins outside the ellipse could not be confidently clustered to a particular serotype as they bound to most of the strains. Blue lines represent strongly correlating proteins ($r^2 > 0.9$), while red line lines represent mutual exclusive proteins or negatively correlating proteins ($r^2 < -0.6$). Thickness of the line represents how strongly the proteins are correlated." We also added ellipse in L.454.

6. L. 512, "no structural model for any E type M interspecies protein complex, we selected M28 for further structural characterization with a particular focus on the binding with IgA and C4BP as outlined in Figure 1B." This is inaccurate as there is an X-ray crystal structure of M28 protein bound to C4BP.

We agree with the reviewer on this point. We have re-phrased the statement (L. 516) to "To visualize how these short E type M proteins form interspecies complex, we selected M28 for further structural characterization with a particular focus on the binding with IgA and C4BP as outlined in Figure 1B.

7. L. 519, "We immobilized the M proteins (ligand) on the sensor chip and injected IgA (analyte) over them to mimic the M proteins protruding out from the bacterial surface and the immunoglobulins floating in the plasma." This is an overstatement, as the M proteins were attached through random amine coupling.

We agree and we have removed the statement "to mimic the M proteins protruding out from the bacterial surface and the immunoglobulins floating in the plasma".

8. The authors should note that the interaction between M28 protein and C4BP is likely to be characterized by avidity, as there are 7 α chains for C4BP and M28 protein is a dimer.

Thank you for the suggestion. We have added a statement (L. 533) stating "As C4BP has 7 α chains and M28 is a dimer so in this case it is likely that we characterize the interaction in terms of avidity. The kinetic analysis for this interaction showed a better fitting to a 1-1 model".

9. L. 549, "These AP-MS experiments were performed using 100% saliva, 1% plasma, 10% plasma in saliva and 100% plasma to mimic conditions during a local infection followed by a systemic infection." This is confusing. It makes it appear that the 1% plasma sample had no saliva.

Sorry for the confusion. We have formatted the sentences L.551 and L.552. Hope this makes the statement clear.

10. Figure 3C. "SF370" is used instead of "M1" and the additional labeling of M28 and M89 require explanation.

Thank you for pointing out. SF370, M3, M28-BB8, M49 and M89-BB4 were mistakenly added to the Figure 3C. It was an error from our end. We have removed those labels as we feel those are redundant; as the M proteins are already marked in the network.

11. L. 638, "A notion that requires further investigations." This sentence is ungrammatical; it lacks a verb.

We have removed the statement from the manuscript.

12. In the Methods, in some places, a space between numbers and units is missing and, in some places, degree signs are missing.

Indeed, at certain occasions a space between the number and the unit was missing and for one of the temperature the degree sign was missing. We have fixed it in the manuscript.

13. L. 130, Values in rpm depend on the rotor. Instead, rcf values should be reported.

Thank you for the comment. We reported all the centrifugation speed in g and we missed three. Those three rpm values have now been changed to g (L.129, L. 202, L.204). Rest of the rpm values are for the shaker. For proper understanding, we have added the shaking information beside these rpm values in L.125, L.179, L.184, L.186, L.193, L. 198, L.208, L.212, L.214, L.224

14. L. 190, "For affinity purification (AP) reactions 20 µg of recombinant affinity-tagged M proteins was charged on Strep-Tactin Sepharose beads (IBA) equilibrated in 1x PBS." What does "charged" mean?

We have modified the statement in L.190 to "For affinity purification (AP) reactions 20 µg of recombinant affinity-tagged M proteins was incubated with 150 µl 50% Strep-Tactin Sepharose beads slurry (IBA) equilibrated in 1x PBS." We also corrected the protein-charged charged term to protein-bound in L.194 and L.197

Reviewer #3 (Comments for the Author):

The elegant study by Chowdhury et al. presents the interaction networks of group A streptococcal M-proteins in plasma (and saliva). The authors show that the interactions depend on (i) GAS serotype and (ii) particularly M protein domain arrangement. Diverse range of network analyses showed that plasma proteins albumin, IgG1, IgG4, and IgA2 are associated with all tested M proteins. In contrast, C4BP α , IgA1, alpha-1-antitrypsin (A1AT), and PROS were mainly associated with M28 and IgA2 showed a significant enrichment with M28. Since the interaction networks of M28 showed such a unique pattern and emm pattern E stains are not well characterized, the authors focused in the second half of the manuscript mainly on M28-interactions. Major conclusions of the study are: (i) M28 binds secretory/polymeric IgA (from saliva) and due to a potential vascular leakage, this binding can be replaced by (ii) monomeric IgA and/or C4BP from plasma. Based on these results the authors conclude that GAS adapt to specific microenvironments by forming protein complexes with host proteins, which in turn helps them to avoid immune detection at the border of mucosal and vascular compartments.

Overall, the study is well designed and performed, clearly presented, and of high clinical relevance. Although the authors used mainly MS-based approaches, major findings, e.g., M28-C4BP and M28-IgA interactions, are also validated and even quantified (calculated K_d-values). I have just a few minor comments.

Minor comments:

1. Lines 60-67: To my knowledge, the classification of emm patterns A-E is not solely based on M protein sequences (domain arrangement). This classification also includes the chromosomal arrangement of emm and emm-like genes (such as enn and mrp). Emm patterns B and C are rare and grouped together with pattern A strains, due to their structural similarities (i.e., all have an SF-1 emm gene and lack mrp). Pattern D and E strains have mrp and the SF-3 form of enn, but are distinct in that their emm genes are of the SF-1 and SF-2 forms, respectively.

Thank you for such a valuable feedback. We agree that M protein classification is not only driven by the protein sequences but it also takes in to account the arrangement and presence of emm and emm-like genes. Therefore, we have updated our statement in L. 60 of the manuscript; "Based on the domain arrangement of the M proteins and the presence of emm and emm-like genes in the GAS genome M proteins are classified."

2. Figure 4A. It would be nice if the KD values would be included within the respective graph.

Yes, we totally agree with the reviewer and thus have added the KD values in the respective graphs.

3. Figures S3 (and 3A)/lines 521-533: The authors tested two fitting models and conclude that heterogeneous ligand model gave the best fit for M28-IgA and M1-IgA interactions, while C4BP-M28 interaction showed a better fitting in a 1-1 model. Just by looking at the Figure S3, I would totally agree with such conclusion concerning M28-IgA interaction. However, I do not see a difference between the fitting models in case of Fig. S3B and C. Could the authors please provide the Chi2 values (measure of the accuracy of the fitting) within the respective graphs, which were probably used to judge which fitting was the best.

In our revised manuscript, we add a table showing the χ^2 values of each interactions and the fitted models in Supplementry figure -3 (S3-D). As you see in the table, for M28-IgA and M1-IgA interactions, the χ^2 values of heterogeneous binding models are lower than the ones for 1-1 binding models. On the other hand, for M28-C4BP binding, 1-1 binding model gives the lowest χ^2 value. These values are consistent with our findings and statistically support the interaction models that we propose in our manuscript.

4. As the authors state in the introduction, binding of M proteins to IgA is believed to prevent IgA-effector functions and inhibits phagocytosis thereby promoting bacterial virulence. However, (i) not every throat infection results in severe infections, (ii) emm28 strains account for a smaller proportion of severe infections (usually among the top 5), and (iii) secretory IgA is rather a weak opsonin in general. Could the authors please discuss how the identified binding sites (M28 and IgA/C4BP) and the resulting binding would impair the function of C4BP and IgA and therefore contribute to the virulence of GAS.

*It is correct that not all throat infections result in severe infections. But as IgA is the most prevalent antibody circulating in the mucosal lining therefore developing a strategy to combat with IgA will be beneficial for the pathogen. For the localised infections *S. pyogenes* needs to bind to mucosal surface, so we think binding of secretory IgA might be involved in adhesion of the bacterium to the cell surface. However, we report that M28 binds to C4BP and IgA in blood. Macrophages are known to express Fc α receptor that bind IgA-Fc. Therefore, binding of IgA-Fc to M28 would inhibit the phagocytosis of the bacterium by macrophages. Moreover, C4BP binding to the strain would prevent the activation of complement pathway therefore bacterial clearance would be inhibited; and there won't be any release of C5a and C3a thereby preventing anaphylaxis. We have included the explanation in the discussion L.664 and L.691.*

August 8, 2021

Dr. Johan Malmström
Lund University
Lund
Sweden

Re: mSystems00271-21R1 (Streptococcus pyogenes forms serotype and local environment-dependent inter-species protein complexes)

Dear Dr. Johan Malmström:

Thank you for submitting your manuscript to mSystems. We have completed our review and I am pleased to inform you that, in principle, we expect to accept it for publication in mSystems. However, acceptance will not be final until you have adequately addressed the reviewer comments.

Preparing Revision Guidelines

For complete guidelines on revision requirements for your article type, please see the journal Article Types requirement at <https://journals.asm.org/journal/mSystems/article-types>. **Submissions of a paper that does not conform to mSystems guidelines will delay acceptance of your manuscript.**

Sincerely,

Frank Schmidt

Editor, mSystems

Journals Department
Reviewer comments:

Reviewer #1 (Comments for the Author):

This reviewer appreciates the authors' responsiveness to the prior critiques. The revision by Chowdhury et al. has largely addressed these prior critiques. That said, the authors should note that there are several discrepancies or unclear points in the revision that are likely to be confusing to readers.

1. "Our data suggests that once an interaction has been established with a particular protein, other plasma protein interactions are not readily formed to the same emm type. Possibly since some of the interacting proteins participate in similar immune evasion functions." What specific interactions does this refer to? Are the authors referring to fibrinogen and Protein S? If so, this is quite far-fetched, as there is no evidence supporting anti-coagulatory roles in GAS infection for either fibrinogen or Protein S. Fibrinogen has been shown to block opsonophagocytic killing, but there is no equivalent evidence for Protein S (or indeed any evidence that Protein S impacts GAS virulence). This paragraph lacks evidentiary support. In addition, the second sentence quoted above is ungrammatical.
2. "Additional AP-MS 622 experiments confirmed that this site is sufficient for enriching IgA but to a lower extent than the SAP sequence (Fig. S5), indicating that this is a low affinity site as suggested by the SPR data (Fig 4A)." This is a problematic statement, because while Fig. S5b shows statistically significant binding of M89 to IgA from human plasma, as compared to the negative control, GFP, Fig. S5c shows that the same binding event is statistically non-significant in the case of commercially acquired IgA. This weakens the contention that the putative additional IgA-binding site found in M28 and M89 is sufficient for interaction. The authors should deal with these less than convincing results.
3. M49 protein is known to bind C4BP. Yet, the experiment with purified proteins in Fig. 4B shows very poor C4BP binding by M49 protein. In contrast, the experiment in Fig. 3A with whole bacteria shows convincing binding between M49 strains and C4BP. The authors should comment on this. If there are other discrepancies between the whole bacteria vs purified protein data, the authors should comment on these as well.
4. Regarding the SPR data, the χ^2 values for the M28-C4BP data are surprisingly high. For the M28-

IgA data, this reviewer wonders if the χ^2 would be better if >2 binding sites were invoked.
5. The authors refer to IgA1 and IgA2, which presumably means monomeric and dimeric (secretory) IgA, respectively. However, this terminology is confusing given that there are IgA1 and IgA2 subclasses.

Reviewer #3 (Comments for the Author):

I would like to thank the authors for adequately addressing my comments. As previously stated, it is an elegant study, which is well designed and performed.

Below we provide the point-by-point responses to the reviewers. For your easy understanding the reviewer comments have been numbered and our response to the reviewer's comments are in italics.

Reviewer #1 (Comments for the Author):

This reviewer appreciates the authors' responsiveness to the prior critiques. The revision by Chowdhury et al. has largely addressed these prior critiques. That said, the authors should note that there are several discrepancies or unclear points in the revision that are likely to be confusing to readers.

1. "Our data suggests that once an interaction has been established with a particular protein, other plasma protein interactions are not readily formed to the same emm type. Possibly since some of the interacting proteins participate in similar immune evasion functions." What specific interactions does this refer to? Are the authors referring to fibrinogen and Protein S? If so, this is quite far-fetched, as there is no evidence supporting anti-coagulatory roles in GAS infection for either fibrinogen or Protein S. Fibrinogen has been shown to block opsonophagocytic killing, but there is no equivalent evidence for Protein S (or indeed any evidence that Protein S impacts GAS virulence). This paragraph lacks evidentiary support. In addition, the second sentence quoted above is ungrammatical.

We agree that this sentence is confusing. We have removed it from the text. For clarity, we have also made minor changes to L.643.

2. "Additional AP-MS 622 experiments confirmed that this site is sufficient for enriching IgA but to a lower extent than the SAP sequence (Fig. S5), indicating that this is a low affinity site as suggested by the SPR data (Fig 4A)." This is a problematic statement, because while Fig. S5b shows statistically significant binding of M89 to IgA from human plasma, as compared to the negative control, GFP, Fig. S5c shows that the same binding event is statistically non-significant in the case of commercially acquired IgA. This weakens the contention that the putative additional IgA-binding site found in M28 and M89 is sufficient for interaction. The authors should deal with these less than convincing results.

One of the differences between the pulldown experiments performed on plasma and commercial IgA was the amount of IgA used for the pulldowns. Consequently, we repeated this experiment one more time and updated Figure S5B and S5C. In this experiment, there is a statistically significant enrichment of commercial IgA to M89, although the levels are significantly lower compared to M28.

3. M49 protein is known to bind C4BP. Yet, the experiment with purified proteins in Fig. 4B shows very poor C4BP binding by M49 protein. In contrast, the experiment in Fig. 3A with whole bacteria shows convincing binding between M49 strains and C4BP. The authors should comment on this. If there are other discrepancies between the whole bacteria vs purified protein data, the authors should comment on these as well.

We guess there has been a miss-numbering with respect to the figures. We think that the reviewer here refers to figure 3B. Figure 3B is a heatmap and it is based on the z-score of the protein intensities. This visualization highlights the most pronounced interactions but makes

it more difficult to judge whether the binding is poor as it the visualization compares the level of C4BPA binding across all M proteins. Fig S2 is more suitable for viewing the enrichment factor for the plasma protein interactors to each M-protein. Fig S2-C shows that C4BPA is statistically enriched on M49 when compared to GFP (>two-fold, corrected p-value <0.05, L. 482) which is in accordance with the findings with whole bacteria shown in Fig 2A

4. Regarding the SPR data, the χ^2 values for the M28-C4BP data are surprisingly high. For the M28-IgA data, this reviewer wonders if the χ^2 would be better if >2 binding sites were invoked.

Chi squared (χ^2) value are a measure of the closeness of a fit in Biacore kinetic evaluations. It is used to describe the differences from the experimental data of the sensorgram to the fitted curve (1,2). This value should be low in relation to the maximum measured response (3). In other words, a low χ^2 value indicates a good fitting to a model. We show that the χ^2 value between M28-IgA is 6.22 for the heterogeneous fitting model. The same χ^2 value for M28-C4BP interaction for 1-1 binding is 88.1. This means that the low M28-IgA χ^2 value is better than the χ^2 value for M28-C4BP.

References:

1. Steinicke et al., *Analytical Biochemistry*, 2017, 53:94-103
2. GE Healthcare Life Sciences Biacore, *BIAevaluation Software Handbook*, 2008
3. Zhang et al., *Biol. Proced. Online*, 2003, 5(1): 170-181.

5. The authors refer to IgA1 and IgA2, which presumably means monomeric and dimeric (secretory) IgA, respectively. However, this terminology is confusing given that there are IgA1 and IgA2 subclasses.

In the manuscript, we use IgA1 and IgA2 to refer to the IgA subclasses (i.e. IgA1 and IgA2). The monomeric form of IgA is denoted as monomeric IgA (L. 510, L. 550, L. 562, L. 577, L. 666, L. 683, L. 686, L. 693, L. 697, L. 699, L. 833) and the dimeric form is denoted as dimeric IgA (L. 617, L. 675, L. 699). Secretory IgA is denoted as either secretory IgA (L. 31, L. 33, L. 548, L. 572, L. 574, L. 577, L. 618, L. 666, L. 668, L. 673, L. 697, L. 833) or sIgA (L. 548, L. 562, L. 565, L. 617, L. 619, L. 620, L. 621, L. 626).

September 6, 2021

Dr. Johan Malmström
Lund University
Lund
Sweden

Re: mSystems00271-21R2 (Streptococcus pyogenes forms serotype and local environment-dependent inter-species protein complexes)

Dear Dr. Johan Malmström:

Your manuscript has been accepted, and I am forwarding it to the ASM Journals Department for publication. For your reference, ASM Journals' address is given below. Before it can be scheduled for publication, your manuscript will be checked by the mSystems senior production editor, Ellie Ghatineh, to make sure that all elements meet the technical requirements for publication. She will contact you if anything needs to be revised before copyediting and production can begin. Otherwise, you will be notified when your proofs are ready to be viewed.

As an open-access publication, mSystems receives no financial support from paid subscriptions and depends on authors' prompt payment of publication fees as soon as their articles are accepted. =

Publication Fees:

We recognize that the video files can become quite large, and so to avoid quality loss ASM suggests sending the video file via <https://www.wetransfer.com/>. When you have a final version of the video and the still ready to share, please send it to Ellie Ghatineh at eghatineh@asmusa.org.

Sincerely,

Frank Schmidt
Editor, mSystems

Journals Department
Supplemental Material: Accept
Fig S4: Accept
Supplemental Material: Accept
Supplemental Material: Accept
Table: Accept
Supplemental Material: Accept
Supplemental Figure 5: Accept
Supplemental Material: Accept